# ARCHITECTURE-AGNOSTIC MASKED IMAGE MODELING – FROM VIT BACK TO CNN

## ABSTRACT

Masked image modeling (MIM), an emerging self-supervised pre-training method, has shown impressive success across numerous downstream vision tasks with Vision transformers (ViTs). Its underlying idea is simple: a portion of the input image is randomly masked out and then reconstructed via the pre-text task. However, the working principle behind MIM is not well explained, and previous studies insist that MIM primarily works for the Transformer family but is incompatible with CNNs. In this paper, we first study interactions among patches to understand what knowledge is learned and how it is acquired via the MIM task. We observe that MIM essentially teaches the model to learn better middle-order interactions among patches and extract more generalized features. Based on this fact, we propose an Architecture-Agnostic Masked Image Modeling framework ($A^2$MIM), which is compatible with both Transformers and CNNs in a unified way. Extensive experiments on popular benchmarks show that our $A^2$MIM learns better representations without explicit design and endows the backbone model with the stronger capability to transfer to various downstream tasks for both Transformers and CNNs.

## 1 INTRODUCTION

Supervised deep learning with large-scale annotated data has witnessed an explosion of success in computer vision (CV) (Krizhevsky et al., 2012a; He et al., 2016) and natural language processing (NLP) (Vaswani et al., 2017). However, a large number of high-quality annotations are not always available in real-world applications. Learning representations without supervision by leveraging pre-text tasks has become increasingly popular.

In CV, early self-supervised learning approaches (Zhang et al., 2016; Doersch et al., 2015; Gidaris et al., 2018) aim to capture invariant features through predicting transformations applied to the same image. However, these methods rely on vision ad-hoc heuristics, and the learned representations are less generic for downstream tasks. Recently, contrastive learning-based approaches (Tian et al., 2020; Chen et al., 2020b; He et al., 2020) have witnessed significant progress, even outperforming supervised methods on several downstream tasks (Chen et al., 2020c; Grill et al., 2020; Zbontar et al., 2021). More recently, inspired by masked autoencoding methods (Radford et al., 2018; Devlin et al., 2018) in NLP, Masked Image Modeling (MIM) methods (Bao et al., 2022; He et al., 2022; Wei et al., 2021; Xie et al., 2021b) have brought about new advances for self-supervised pre-training on CV tasks. The transition from human language understanding to NLP masked autoencoding is quite natural because the filling of missing words in a sentence requires relatively comprehensive semantic understanding. In analogy, humans can understand and imagine masked content by visually filling the missing structures in an image containing occluded parts.

Different from contrastive learning, which yields a clustering effect from pre-training by pulling similar samples and pushing away dissimilar samples, MIM pre-training methods have not been extensively explored in the context of the expected knowledge learned or how this knowledge is acquired. Existing works mainly focus on improving downstream tasks performance via explicit design such as trying different prediction target (Wei et al., 2021), adopting pre-trained tokenizer (Zhou et al., 2021), utilizing complex Transformer decoder (He et al., 2022) or combining with contrastive learning (El-Nouby et al., 2021). Moreover, the success of existing MIM methods is largely confined to Vision Transformer (ViT) structures (Dosovitskiy et al., 2021) since it leads to under-performing performance to directly apply mask token (Devlin et al., 2018) and positional embedding to CNNs.

In this work, we carry out systematic experiments and show that MIM as a pre-training task essentially teaches the model to learn better middle-order interactions between patches for more generalized feature extraction regardless of the underlying network structure. Compared to the local texture features learned by low-order interactions between patches, more complex features such as shape and edge could be extracted via middle-order interactions among patches. The interaction of patches could be considered as information fusion via both the convolution operation of a CNN and the self-attention mechanism of a Transformer. That is to say, CNN and Transformer should both benefit from better middle-order interactions with MIM as the pre-text task.

To bridge the gap of MIM in terms of network architectures based on our extensive experimental analysis, we propose an Architecture-Agnostic Masked Image Modeling framework ($A^2$MIM) that focuses on enhancing the middle-order interaction capabilities of the network. Specifically, we mask the input image with the mean RGB value and place the mask token at intermediate feature maps of the network. In addition, we propose a loss in the Fourier domain to further enhance the middle-order interaction capability of the network. Our contributions are summarized as follows:

- We conducted systematic experiments and showed the essence of MIM is to better learn middle-order interactions between patches but not reconstruction quality.
- We proposed a novel MIM-based framework dubbed $A^2$MIM that bridges the gap between CNNs and Transformers. We are also the first to perform MIM on CNNs without adopting designs native to ViTs that outperforms contrastive learning counterparts.
- Extensive experiments with both Transformers and CNNs on ImageNet-1K and public benchmarks for various downstream tasks show that our method achieves performance improvement on pre-trained representation quality than state-of-the-art methods.

## 2 RELATED WORK

**Contrastive Learning.** Contrastive learning learns instance-level discriminative representations by extracting invariant features over distorted views of the same data. MoCo (He et al., 2020) and SimCLR (Chen et al., 2020b) adopted different mechanisms to introduce negative samples for contrast with the positive. BYOL (Grill et al., 2020) and its variants (Chen & He, 2020; Chen et al., 2021) further eliminate the requirement of negative samples to avoid representation collapse. Besides pairwise contrasting, SwAV (Caron et al., 2020) clusters the data while enforcing consistency between multi-augmented views of the same image. Barlow Twins (Zbontar et al., 2021) proposed to measure the cross-correlation matrix of distorted views of the same image to avoid representation collapsing. Meanwhile, some efforts have been made on top of contrastive methods to improve pre-training quality for specific downstream tasks (Xie et al., 2021a; Xiao et al., 2021; Selvaraju et al., 2021; Wu et al., 2022). MoCo.V3 (Chen et al., 2021) and DINO (Caron et al., 2021) adopted ViT (Dosovitskiy et al., 2021) in self-supervised pre-training to replace CNN backbones.

**Autoregressive Modeling.** Autoencoders (AE) are a typical type of network architecture that allows representation learning with no annotation requirement (Hinton & Zemel, 1993). By forcing denoising property onto the learned representations, denoising autoencoders (Vincent et al., 2008; 2010) are a family of AEs that reconstruct the uncorrected input signal with a corrupted version of the signal as input. Generalizing the notion of denoising autoregressive modeling, masked predictions attracted the attention of both the NLP and CV communities. BERT (Devlin et al., 2018) performs masked language modeling (MLM) where the task is to classify the randomly masked input tokens. Representations learned by BERT as pre-training generalize well to various downstream tasks. For CV, inpainting tasks (Pathak et al., 2016) to predict large missing regions using CNN encoders and colorization tasks (Zhang et al., 2016) to reconstruct the original color of images with removed color channels are proposed to learn representation without supervision. With the introduction of Vision Transformers (ViT) (Dosovitskiy et al., 2021; Liu et al., 2021), iGPT (Chen et al., 2020a) predicts succeeding pixels given a sequence of pixels as input. MAE (He et al., 2022) and BEiT (Bao et al., 2022) randomly mask out input image patches and reconstruct the missing patches with ViTs. Compared to MAE, MaskFeat (Wei et al., 2021) and SimMIM (Xie et al., 2021b) adopt linear layers as the decoder instead of another Transformer as in MAE. MaskFeat applied HOG as the prediction target instead of the RGB value. Other research endeavors (El-Nouby et al., 2021; Zhou et al., 2021; Assran et al., 2022; Akbari et al., 2021; Sameni et al., 2022) combine the idea of contrastive learning

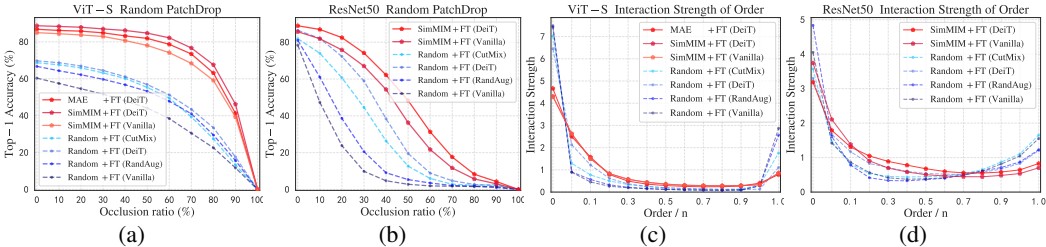

Figure 1: (a)(b): Robustness against different occlusion ratios of images is studied for both ViT-S and ResNet-50 under different experimental settings (see Section 3.1). (c)(d): Distributions of the interaction strength $J^{(m)}$ are explored for both ViT-S and ResNet-50 under different experimental settings. The label indicates the pre-training method + fine-tuning augmentation used, random stands for random weight initialization. Appendix B provides more results and implement details.

(CL) with MIM. SplitMask (El-Nouby et al., 2021) proposed to use half of the image pixels to predict the other half while applying InfoNCE loss (Van den Oord et al., 2018) across the corresponding latent features. MSN (Assran et al., 2022) matches the representation of an image view containing randomly masked patches and the original unmasked image. Similarly, iBOT (Zhou et al., 2021) adopts the Siamese framework to combine self-distillation with MIM. Moreover, Data2Vec (Baevski et al., 2022) proposed a framework that applies the masked prediction idea for either speech, NLP, or CV. However, most MIM works are confined to ViT architectures, recently proposed CIM (Fang et al., 2022) adopts the output of a pre-trained tokenizer as target and takes the prediction of a frozen BEiT as input to the encoder as a workaround to enable MIM on CNNs. In this work, we propose $A^2$MIM with no components native to ViTs adopted to perform MIM with ViTs and CNNs.

# 3 INTRIGUING PROPERTIES OF MASKED IMAGE MODELING

## 3.1 IS MIM BETTER IMAGE AUGMENTATION?

Compared to CNN, Transformers gain tremendous performance improvement with carefully designed image augmentation techniques such as RandAug(Cubuk et al., 2020), CutMix(Yun et al., 2019) and random erasing(Zhong et al., 2020). Random erasing(Zhong et al., 2020) randomly removes part of the image and replaces it with Gaussian noise, while Cutmix randomly removes part of the image and replaces the corresponding region with a patch from another image. Similarly, as in most MIM pre-training tasks, some image patches are masked out and replaced with a learnable mask token. Noticing the resemblance of the masking operations, *we hypothesize that MIM as a pre-training task and masking-based data augmentations enhance the network's robustness towards occlusion, enabling the network with a more generalized feature extraction ability.* To verify our hypothesis, we design an occlusion robustness test. Let $x \in \mathbb{R}^{3 \times H \times W}$ be an input image and $y \in \mathbb{R}^C$ be its corresponding label, where $C$ is the class number. Consider a classification task $y = f(x)$ where $f$ denotes a neural network, the network is considered robust if the network outputs the correct label given an occluded version of the image $x'$, namely $y = f(x')$. For occlusion, we consider the patch-based random masking as adopted in most MIM works (He et al., 2022; Xie et al., 2021b; Wei et al., 2021). In particular, we split the image of size $224 \times 224$ into patch size $16 \times 16$ and randomly mask $M$ patches out of the total number of $N$ patches. The occlusion ratio could then be defined as $\frac{M}{N}$. We conduct experiments on ImageNet-100 (IN-100) (Krizhevsky et al., 2012b) for both Transformer and CNN with different settings. We choose ViT-S (Dosovitskiy et al., 2021) and ResNet-50(He et al., 2016) as the network architecture. Robustness is compared under the following settings: **(i)** random weight initialization with no image augmentation applied; **(ii)** random weight initialization with different image augmentations applied; **(iii)** MIM pre-training as weight initialization with and without image augmentations applied. In Fig. 1, we report the average top-1 accuracy across five runs trained with different settings under various occlusion ratios. Fig. 1(a) and 1(b) show that both MIM and patch-removing alike augmentations significantly improve model occlusion robustness for both ViT-S and ResNet-50. Nevertheless, MIM yields more robust feature extraction than adopting augmentations. Although MIM and patch-removing alike augmentations share similar masking mechanisms, MIM explicitly forces the model to learn the interactions between patches in order to reconstruct missing patches enabling more robust feature extraction. Comparing Fig. 1(a) and 1(b), the convex trend of accuracy from ViT-S indicates better robustness than the concave trend from ResNet-50. The self-attention mechanism of ViTs is able to model the interactions between patches with high degrees of freedom compared to CNNs constrained by convolution priors. *We claim that*

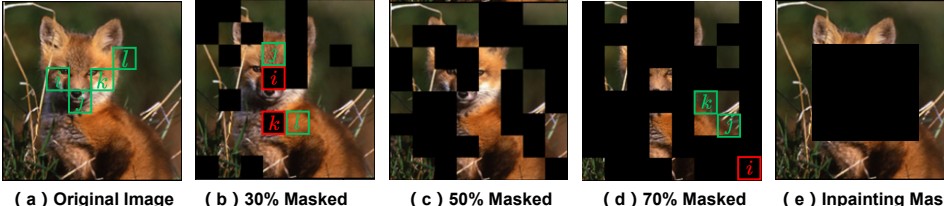

( a ) Original Image    ( b ) 30% Masked    ( c ) 50% Masked    ( d ) 70% Masked    ( e ) Inpainting Mask

Figure 2: (a) Four patches $(i, j, k, l)$ interact with each other and forms a contour or edge pattern of the fox for image categorization. (b) Image with 30% masking ratio. Masked patches $i$ and $k$ interact with neighboring patches $j$ and $l$ to predict the missing patches. (c) Image with 50% masking ratio. Masked patches are forced to interact with middle-order interactions for the MIM task. (d) Image with 70% masking ratio. Masked Patch $i$ interacts with longer-range patches $j$ and $k$, forming an edge pattern. (e) A typical masking pattern for existing inpainting tasks.

*the success of MIM on ViTs can be seen as resonance in terms of better patch interactions imposed by MIM while supported by the self-attention mechanism of ViTs.*

## 3.2 Middle-order Interactions for Generalized Feature Extraction

Next, we show that MIM essentially enables better middle-order interactions between patches. Note that existing MIM works adopt a medium or high masking ratio (Xie et al., 2021b; He et al., 2022) (*e.g.*, 60% or 70%, see Fig. 2) during pre-training, and in these settings, the pairwise interactions between patches are under a middle-size context measured by the order $m$. Early inpainting work based on CNN (Pathak et al., 2016) resembles MIM but attracts little attention due to the much inferior performance to contrastive learning methods. The inpainting task adopts the masking strategy as illustrated in Fig. 1(c), which masks a full large region instead of random small patches. Such masking mechanisms ignore patch interaction and focus only on reconstruction leading to poor, learned representation quality. To investigate whether MIM makes the model more sensitive to patch interactions of some particular orders, we resort to the tool of multi-order interactions introduced by (Deng et al., 2022; Zhang et al., 2020). Intuitively, $m^{th}$-order interactions of patches refer to inference patterns (deep features) induced from $m$ number of patches of the original image in the input space. With a small value of $m$ (low-order interactions), the model simply learns local features such as texture. Formally, the multi-order interaction $I^{(m)}(i, j)$ is to measure the order of interactions between patches $i$ and $j$. We define $I^{(m)}(i, j)$ to be the average interaction utility between patches $i$ and $j$ on all contexts consisting of $m$ patches. $m$ indicates the order of contextual complexity of the interaction. Mathematically, given an input image $x$ with a set of $n$ patches $N = \{1, \ldots, n\}$ (*e.g.*, an image with $n$ pixels), the multi-order interaction $I^{(m)}(i, j)$ is defined as:

$$I^{(m)}(i, j) = \mathbb{E}_{S \subseteq N \setminus \{i,j\}, |S|=m}[\Delta f(i, j, S)], \tag{1}$$

where $\Delta f(i, j, S) = f(S \cup \{i, j\}) - f(S \cup \{i\}) - f(S \cup \{j\}) + f(S)$. $f(S)$ indicates the score of output with patches in $N \setminus S$ kept unchanged but replaced with the baseline value (Ancona et al., 2019), where the context $S \subseteq N$. See Appendix B.2 for details. To measure the interaction complexity of the neural network, we measure the relative interaction strength $J^{(m)}$ of the encoded $m$-th order interaction as follow:

$$J^{(m)} = \frac{\mathbb{E}_{x \in \Omega} \mathbb{E}_{i,j} |I^{(m)}(i, j|x)|}{\mathbb{E}_{m'} \mathbb{E}_{x \in \Omega} \mathbb{E}_{i,j} |I^{(m')}(i, j|x)|}, \tag{2}$$

where $\Omega$ is the set of all samples and $0 \le m \ge n - 2$. $J^{(m)}$ is the average value over all possible pairs of patches of input samples. $J^{(m)}$ is normalized by the average value of all interaction strengths. $J^{(m)}$ then indicates the distribution (area under curve sums up to one) of the order of interactions of the network. In this work, we use $J^{(m)}$ as the metric to evaluate and analyze interaction orders of the network with MIM pre-training. We conduct experiments on IN-100 with image size $224 \times 224$ and use ViT-S (Dosovitskiy et al., 2021) and ResNet-50 (He et al., 2016) as the network architecture. We consider a patch of size $16 \times 16$ as an input patch. For the computation of $J^{(m)}$, we adopt the sampling solution following previous works (Deng et al., 2022; Zhang et al., 2020). As can be seen from Fig. 1(c) that ViT-S with random weight initialization tends to learn simple interactions with few patches (e.g., less than $0.05n$ patches) while MIM pre-trained models show a stronger interaction for relative middle-order (from $0.05n$ to $0.5n$). Similarly, as observed from 1(d), MIM pre-trained

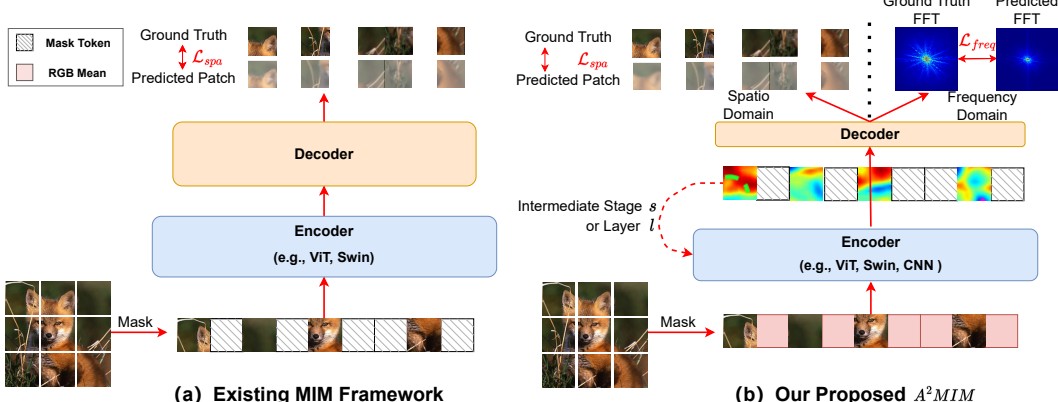

(a) Existing MIM Framework      (b) Our Proposed $A^2MIM$

Figure 3: An illustration comparison between the existing MIM framework and our proposed framework. For the existing MIM framework, the input image is patchfied into a sequence of patches without overlapping with masked patches that are replaced with learnable mask tokens. The sequence is then input to the Transformer encoder. The $\mathcal{L}_{spa}$ is applied between the ground truth patches and the reconstructed patches from the decoder in the spatiotemporal domain. Our proposed framework uses the mean RGB value of the image instead of the mask token in the input space. We then add a learnable mask token onto the intermediate feature map of layer-$l$ of stage-$s$ of the encoder instead of replacement in the input space. The encoder could either be of the Transformer or the CNN family. In addition to the $\mathcal{L}_{spa}$, we adopt a $\mathcal{L}_{freq}$ in the Fourier domain to enhance the encoder to learn more middle-order interactions. Specifically, we apply DFT on both the ground truth image and the predicted image and then use Mean square error (MSE) to measure the difference.

ResNet-50 enhances the middle-order interactions from $0.1n$ to $0.55n$ compared to random initialized models. Stronger middle-order interactions form more complex features such as shape and edge compared to local texture features learned from low-order interactions (Naseer et al., 2021).

## 4   APPROACH

We propose a generic MIM framework following two design rules: (a) **No complex or non-generic designs are adopted to ensure compatibility with all network architectures.** (b) **Better middle-order interactions between patches for more generalized feature extraction.** Figure 3 highlights the difference between our proposed framework and existing MIM frameworks in terms of three key components: masking strategy, encoder/decoder architecture design and prediction targets.

### 4.1   ARCHITECTURE AGNOSTIC FRAMEWORK

**Mask Where Middle-order Interactions Occur.**   Existing works (El-Nouby et al., 2021; He et al., 2022; Xie et al., 2021b; Wei et al., 2021) adopt the masking strategy where the input image is divided into non-overlapping patches, and a random subset of patches are masked. MAE utilizes a Transformer as a decoder and takes only the visible patches into the encoder. Masked tokens are appended to the decoder to reconstruct the masked patches. SimMIM (Xie et al., 2021b) and MaskFeat (Wei et al., 2021) utilize a fully connected layer as the decoder and feed the mask token into the encoder together with the visible patches. The mask token (Devlin et al., 2018) is a token-shared learnable parameter that indicates the presence of missing patches to be predicted. Despite different choices of decoder structures, the mask token is either placed at the input to the encoder or the decoder. Mathematically, the masking process of MIM is defined as $x_{mask} = x \odot (1 - M) + T \odot M$, where $M$ is the random occlusion mask, and $T$ represents the learnable mask token. Such masking at the patch embedding layer aligns with the attention mechanism of Transformers, which is robust against occlusion. However, masking at the stem layer undermines the context extraction capability of CNN, which relies on local inductive biases. Moreover, masking at input stages of the network leads to low-order interactions. Thus, we propose to mask intermediate features where the output feature contains both the semantic and spatial information and the mask token can encode interactions with the medium number of tokens. More concretely, our masking operation is defined as $z_{mask}^l = z^l + T \odot D(M)$, where $z^l$ is the intermediate feature map of $x$ at layer-$l$ in the Transformer encoder (or for stage-$l$ in CNNs) and $D(\cdot)$ is the corresponding down-sampling function of the occlusion mask.

**Filling Masked Tokens with RGB Mean.** It is worth noting that existing works directly replace the occluded patches with the mask token in the input space or after the patch embedding (Bao et al., 2022; Xie et al., 2021b). In contrast, we use the average RGB value to fill the occluded patches as the input to the encoder and add the mask token onto the intermediate feature maps of the encoder. The masking mechanism originates from NLP where languages are of high-level semantics and do not require low-level feature extraction as image processing. The introduction of a zero mask at the early stages of the network where low-level feature extraction happens is harmful in terms of feature extraction. From the view of Fourier domain, the RGB mean is the DC component of images. It not only brings about minimum local statistics variation caused by the masking operation but also forces the network to model rather more informative medium frequencies instead of filling the mask patches with blurry color blocks of low frequencies. The proposed masking strategy is generic to both convolution and self-attention in that it accommodates low-level to semantic-level feature extraction.

## 4.2 MIDDLE-ORDER INTERACTIONS FROM FOURIER PERSPECTIVE

Current works (El-Nouby et al., 2021; He et al., 2022; Xie et al., 2021b) adopt raw RGB values as the prediction target. However, raw pixels in the spatial domain are heavily redundant and often contain low-order statistics (Bao et al., 2022; Wei et al., 2021; Zhou et al., 2021). MaskFeat (Wei et al., 2021) adopts the Histogram of Oriented Gradients (HOG) as the prediction target outperforming MAE and SimMIM. HOG is a discrete descriptor of medium or high-frequency features which captures shape patterns based on middle-order interactions. ViTs and CNNs have low-pass and high-pass filtering properties, respectively (Park & Kim, 2022; 2021). ViTs and CNNs have certain frequency bands that they each cannot model well, and both cannot model middle-order interactions well (detailed in Appendix B.3). The observation of the medium frequency descriptor HOG improves middle-order interactions and leads to the hypothesis that learning medium frequencies would help the model learn more middle-order interactions. Given a RGB image $x \in \mathbb{R}^{3 \times H \times W}$, the discrete Fourier transform (DFT) of each channel is defined as:

$$F_{(u,v)} = \sum_{h=1}^{h=H} \sum_{w=1}^{w=W} x(h,w) e^{-2\pi j(\frac{uh}{H} + \frac{vw}{W})}. \tag{3}$$

In addition to the common MIM loss in the spatial domain $\mathcal{L}_{spa}$, we propose $\mathcal{L}_{freq}$ in Fourier domain:

$$\mathcal{L}_{freq} = \sum_{c=1}^{c=3} \sum_{u=1}^{u=H} \sum_{w=1}^{w=W} \omega(u,v) \left\| \text{DFT}(x_c^{pred} \odot M + \text{de}(x_c^{pred}) \odot (1-M)) - \text{DFT}(x_c) \right\|, \tag{4}$$

where $x^{pred}$ is the predicted image, $\text{de}(\cdot)$ is `detach` gradient operation, and $\omega(u,v)$ is the frequency weighting matrix. $\omega(u,v)$ enables both ViTs and CNNs to model features of medium frequencies rather than local textures and noise corresponding to high frequencies. Inspired by Focal Frequency loss (Jiang et al., 2021), we define adaptive $\omega(u,v)$ as follows:

$$\omega(u,v) = \left\| \text{DFT}(x_c^{pred} \odot M + \det(x_c^{pred}) \odot (1-M)) - \text{DFT}(x_c) \right\|^{\alpha}, \tag{5}$$

where $\alpha$ is a scaling factor, and we set $\alpha$ = 1. Fig. B.3 verifies that Eq. (5) allows the model to learn previously ignored frequencies (mostly the medium frequency components). Note that $\mathcal{L}_{freq}$ introduces negligible overhead by using Fast Fourier Transform (FFT) algorithms with $\mathcal{O}(n \log n)$ complexity to achieve DFT. The overall loss function of A²MIM is then defined as:

$$\mathcal{L} = \mathcal{L}_{spa} + \lambda \mathcal{L}_{freq}, \tag{6}$$

where $\mathcal{L}_{spa} = \left\| x^{pred} - x \right\| \odot M$ and $\lambda$ is a loss weighting parameter. We set $\lambda$ to 0.5 by default.

## 5 EXPERIMENTS

### 5.1 PRE-TRAINING SETUP

We adopt ResNet-50 (He et al., 2016) and Vision Transformer (Dosovitskiy et al., 2021) (ViT-S/16 and ViT-B/16) as the backbone. We pre-train on ImageNet-1K (IN-1K) training set with AdamW (Loshchilov & Hutter, 2019) optimizer with a basic learning rate of $1.5 \times 10^{-4}$ adjusted by

a cosine learning rate scheduler and a batch size of 2048. The input image size is $224 \times 224$ with a patch size of $32 \times 32$. We use a random masking ratio of 60%. By default, the learnable mask tokens are placed at stage-3 in ResNet-50 and layer-5/layer-8 in ViT-S/ViT-B, respectively. We adopt a linear prediction head as the decoder (Xie et al., 2021b). A$^2$MIM+ indicates adopting HOG as supervision and using the MLP decoder with depth-wise (DW) convolution. Our experiments are implemented on OpenMixup (Li et al., 2022) by Pytorch and conducted on workstations with NVIDIA V100 GPUs. We report the average results of 3 trials for all experiments and use **bold** and underline to indicate the best and the second-best performance. See Appendix A for detailed pre-training settings.

## 5.2 IMAGE CLASSIFICATION ON IMAGENET-1K

**Evaluation Protocols.** We first evaluate the learned representation by end-to-end fine-tuning (FT) and linear probing (Lin.) protocols on IN-1K. For evaluation on CNN, we adopt RSB A2/A3 (Wightman et al., 2021) training settings for fine-tuning on ResNet-50, which employs LAMB (You et al., 2020) optimizer with a cosine scheduler for 300/100 epochs. For the linear probing setting on ResNet-50, we freeze the backbone features and train a linear classifier with an initial learning rate of 30 and batch size of 256 following MoCo (He et al., 2020). For evaluation on Transformer, we employ the fine-tuning as MAE (He et al., 2022), which uses DeiT (Touvron et al., 2021) augmentation setting, an AdamW optimizer for 100-epoch training, and adopt a layer-wise learning rate decay of 0.65 following (Bao et al., 2022). See Appendix A for detailed evaluation configurations.

**ResNet-50.** We compare the proposed A$^2$MIM with classical self-supervised learning methods (Inpainting (Pathak et al., 2016), Relative-Loc (Doersch et al., 2015), and Rotation (Gidaris et al., 2018)), contrastive learning (CL), and MIM methods with 100/300 pre-training epochs. We modified MIM methods to run them on ResNet-50: the learnable mask token is employed to the encoder of BEiT (Bao et al., 2022), Data2Vec (Baevski et al., 2022), and SimMIM (Xie et al., 2021b) after the

Table 1: ImageNet-1K linear probing (Lin.) and fine-tuning (FT) top-1 accuracy (%) of ResNet-50.

†Multi-crop augmentation.    ‡Our modified version for CNN.

| Method | Fast Pre-training | | | Longer Pre-training | | |
|---|---|---|---|---|---|---|
| | Epochs | Lin. | FT (A3) | Epochs | FT (A3) | FT (A2) |
| PyTorch (Sup.) | 90 | 76.6 | 78.8 | 300 | 78.9 | 79.8 |
| Inpainting | 70 | 40.1 | 78.4 | 300 | 78.0 | - |
| Relative-Loc | 70 | 38.8 | 77.8 | 300 | 77.9 | - |
| Rotation | 70 | 48.1 | 77.7 | 300 | 78.2 | - |
| SimCLR | 100 | 64.4 | 78.5 | 800 | 78.8 | 79.9 |
| MoCoV2 | 100 | 66.8 | 78.5 | 800 | 78.8 | 79.8 |
| BYOL | 100 | 68.4 | 78.7 | 400 | 78.9 | 80.1 |
| SwAV† | 100 | 71.9 | **78.9** | 400 | **79.0** | 80.2 |
| Barlow Twins | 100 | 67.2 | 78.5 | 300 | 78.8 | 79.9 |
| MoCoV3 | 100 | 68.9 | 78.7 | 300 | **79.0** | 80.1 |
| BEiT‡ | 100 | 44.6 | 78.1 | - | - | - |
| Data2Vec‡ | 100 | 43.2 | 78.0 | - | - | - |
| MAE‡ | 100 | 37.8 | 77.1 | - | - | - |
| SimMIM‡ | 100 | 47.5 | 78.2 | 300 | 78.1 | 79.7 |
| CIM | - | - | - | 300 | 78.6 | 80.4 |
| **A$^2$MIM** | 100 | 48.1 | 78.8 | 300 | 78.9 | 80.4 |
| **A$^2$MIM+** | 100 | 50.3 | **78.9** | 300 | **79.0** | 80.5 |

Table 2: ImageNet-1K fine-tuning (FT) top-1 accuracy (%) of ViT-S and ViT-B models.

| Method | Supervision | ViT-S | | ViT-B | |
|---|---|---|---|---|---|
| | | Epochs | FT | Epochs | FT |
| DeiT (Sup.) | Label | 300 | 79.9 | 300 | 81.8 |
| MoCoV3 | CL | 300 | 81.4 | 300 | 83.2 |
| DINO | CL | 300 | 81.5 | 400 | 83.6 |
| BEiT | DALLE | 800 | 81.3 | 800 | 83.2 |
| SplitMask | DALLE+CL | 300 | 81.5 | 300 | 83.6 |
| iBOT | Momentum | 800 | 82.3 | 400 | 84.0 |
| MAE | RGB | - | - | 1600 | 83.6 |
| MaskFeat | HoG | - | - | 800 | 84.0 |
| Data2Vec | Momentum | - | - | 800 | 84.2 |
| SimMIM | RGB | 300 | 81.7 | 800 | 83.8 |
| CAE | DALLE | 300 | 81.8 | 800 | 83.6 |
| CIM | BEiT | 300 | 81.6 | 300 | 83.3 |
| *mc*-BEiT | VQGAN | - | - | 800 | 84.1 |
| BootMAE | Momentum | - | - | 800 | 84.2 |
| **A$^2$MIM** | RGB | 300 | 82.2 | 800 | 84.2 |
| **A$^2$MIM+** | HoG | 300 | **82.4** | 800 | **84.5** |

Table 3: Performance of object detection and semantic segmentation tasks based on ResNet-50 on COCO and ADE20K.

| Method | Epochs | COCO | | ADE-20K |
|---|---|---|---|---|
| | | AP$^{box}$ | AP$^{mask}$ | mIoU |
| PyTorch (Sup.) | 120 | 38.2 | 33.3 | 36.1 |
| SimCLR | 800 | 37.9 | 33.3 | 37.6 |
| MoCoV2 | 400 | 39.2 | 34.3 | 37.5 |
| BYOL | 400 | 38.9 | 34.2 | 37.2 |
| SwAV | 800 | 38.4 | 33.8 | 37.3 |
| SimSiam | 400 | 39.2 | 34.4 | 37.2 |
| Balow Twins | 800 | 39.2 | 34.3 | 37.3 |
| SimMIM‡ | 300 | 39.1 | 34.2 | 37.4 |
| CIM | 300 | - | - | 38.0 |
| **A$^2$MIM** | 300 | **39.8** | **34.9** | **38.3** |

Table 4: Performance of object detection and semantic segmentation tasks based on ViT-B on COCO and ADE-20K.

| Method | Supervision | Epochs | COCO | | ADE-20K |
|---|---|---|---|---|---|
| | | | AP$^{box}$ | AP$^{mask}$ | mIoU |
| DeiT (Sup.) | Label | 300 | 47.9 | 42.9 | 47.0 |
| MoCoV3 | CL | 300 | 47.9 | 42.7 | 47.3 |
| DINO | CL | 400 | 46.8 | 41.5 | 47.2 |
| BEiT | DALLE | 300 | 43.1 | 38.2 | 47.1 |
| iBOT | Momentum | 400 | 48.4 | 42.7 | 48.0 |
| MAE | RGB | 1600 | 48.5 | 42.8 | 48.1 |
| MaskFeat | HoG | 800 | 49.2 | 43.2 | 48.8 |
| SimMIM | RGB | 800 | 48.9 | 43.0 | 48.4 |
| CAE | DALLE | 800 | 49.2 | 43.3 | 48.8 |
| **A$^2$MIM** | RGB | 800 | **49.4** | **43.5** | **49.0** |

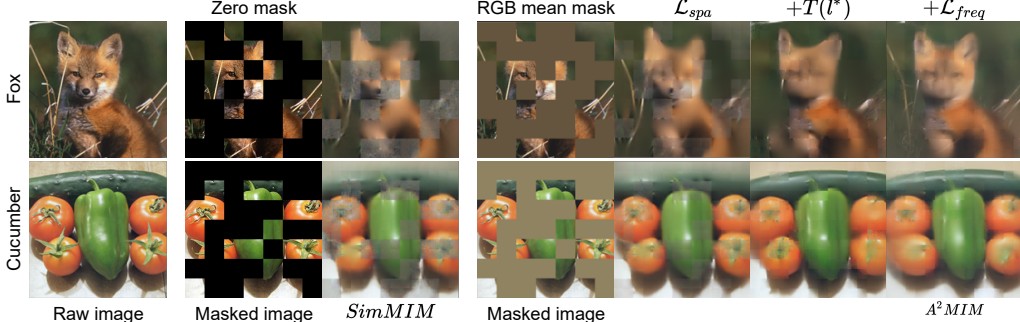

Figure 4: Visualizations of predicted results from SimMIM (middle) and our A$^2$MIM (right) based on ViT-S pre-trained 300-epochs on IN-1K. Notice that $T(l^*)$ denotes the mask token $T$ to the optimal layer-5 in ViT-S. We ablate the proposed components by adding them to the baseline. Compared to results from SimMIM, reconstruction results of the modified baseline ($\mathcal{L}_{spa}$) with the RGB mean mask relieves grid-like artifacts; adding the mask token $T(l^*)$ further improves the smoothness; using the proposed $\mathcal{L}_{freq}$ helps the model to capture more informative details and contours.

stem (the output feature of $56 \times 56$ resolutions); the encoder of MAE randomly selects 25% from $56 \times 56$ output features of the stem as unmasked patches and takes the reorganized $28 \times 28$ patches as the input of four stages. As shown in Tab. 1, our approach achieves competitive performance with state-of-the-art contrastive-based methods under 100-epoch RSB A3 fine-tuning. Note that MIM methods see fewer training samples per epoch than CL methods (40% *vs.* 200% of patches) and usually require longer pre-training epochs. Based on a longer fine-tuning evaluation using RSB A2, our method (300-epoch) outperforms contrastive-based methods with even fewer training epochs. Meanwhile, A$^2$MIM also improves the baseline SimMIM$^\dagger$ (+0.8%) and the concurrent work CIM (+0.4%) in terms of RSB A3 fine-tuning for the longer pre-training. Besides, we also report the linear probing accuracy in the fast pre-training for reference, although our main focus is to learn representations with better fine-tuning performances. The linear probing performance of our method is lower than contrastive-based methods, it still improves the baseline by 0.6%.

**ViT.** We then evaluate A$^2$MIM based on ViT-S/B in Tab. 2. We list the supervision target used by various pre-training methods in the second column of Tab. 2. DALL-E (Ramesh et al., 2021) and VQGAN (Esser et al., 2021) are pre-trained image tokenizers, while momentum refers to the momentum encoder. Our approach outperforms current state-of-the-art methods with complex supervision, *e.g.,* SplitMask (MIM with CL combined), iBOT (complex teacher-student architecture), and CIM (pre-trained BEiT as supervision). Based on ViT-S/B, A$^2$MIM improves the baseline SimMIM by 0.5%/0.4% with RGB as supervision and 0.7%/0.7% with the HOG feature as supervision.

## 5.3 TRANSFER LEARNING EXPERIMENTS

**Object detection and segmentation on COCO.** To verify the transferring abilities, we benchmark CL and MIM methods on object detection and segmentation with COCO (Lin et al., 2014). For evaluation on CNN, we follow the setup in MoCo, which fine-tunes Mask R-CNN (He et al., 2017) with ResNet-50-C4 backbone using 2× schedule on the COCO *train2017* and evaluates on the COCO *val2017*. Results in Tab. 3 indicate that our approach (300-epoch) outperforms contrastive-based methods with longer pre-training (+0.7% AP$^{box}$ and +0.6% AP$^{mask}$). For evaluation on Transformer, we follow MAE and CAE, which efficiently fine-tunes Mask R-CNN with ViT-B backbone using 1× schedule. In Tab. 4, our approach (800-epoch) is superior to popular contrastive-based and MIM methods, *e.g.*, outperforms MAE (1600-epoch) by 0.9% AP$^{box}$ and 0.8% AP$^{mask}$.

**Semantic segmentation on ADE20K.** We then evaluate the transferring performances on semantic segmentation with ADE20K (Zhou et al., 2019) by fine-tuning UperNet (Xiao et al., 2018). Based on ResNet-50, all CNN models are fine-tuned for 160K iterations with SGD following MoCo. Results in Tab. 3 show that our method outperforms CL methods by at least 0.9% mIoU and outperforms CIM (required extra pre-trained BEiT (Bao et al., 2022)) by 0.3% mIoU. Based on ViT-B, we fine-tune models for 80K iterations with AdamW following MAE. Tab. 4 shows that our approach consistently improves MIM methods (*e.g.,* outperforms MAE and SimMIM by 0.9% and 0.6% mIoU).

Figure 5: Ablation of mask token in various stages (S) or layers (L) based on SimMIM (without $\mathcal{L}_{freq}$) on IN-100.

Table 5: Ablation of A$^2$MIM on IN-100 and IN-1K. w/o $\omega$ denotes removing the re-weighting term $\omega$ in $\mathcal{L}_{freq}$ and $T(l^*)$ denotes adding the mask token $T$ to the optimal layer-$l^*$.

| Backbones | ResNet-50 | | ViT-S | ViT-B |
|---|---|---|---|---|
| Datasets | IN-100 | IN-1K | IN-100 | IN-1K |
| PT Epochs | 400 ep | 100 ep | 400 ep | 400 ep |
| SimMIM | 87.75 | 78.2 | 85.10 | 83.1 |
| $\mathcal{L}_{spa}$ | 88.19 | 78.4 | 85.27 | 83.2 |
| $+\mathcal{L}_{freq}$ w/o $\omega$ | 88.47 | 78.4 | 86.05 | 83.3 |
| $+\mathcal{L}_{freq}$ | 88.73 | 78.6 | 86.41 | 83.4 |
| $+\mathcal{L}_{freq} + T(l^*)$ | **88.86** | **78.8** | **86.62** | **83.5** |

## 5.4 ABLATION STUDY

We next verify the effectiveness of the proposed components. Ablation studies are conducted with ResNet-50 and ViTs on IN-100 and IN-1K using the fine-tuning protocol. Based on the modified baseline SimMIM ($\mathcal{L}_{spa}$), we first compare different mask token mechanisms: **Replacing** denotes the original way in most MIM methods, and **Addition** denotes our proposed way that adds the mask token to intermediate feature maps of the backbone. As shown in Fig. 5, adding the mask token to the medium stages (stage-3) or layers (layer-5) yields the best performance. Replacing masked patches in input images by RGB mean value slightly improves the baseline SimMIM, especially for ResNet-50 (88.19 *vs.* 87.75 on IN-100). Then, we verify the proposed $\mathcal{L}_{freq}$ in Tab. 5. We find that simply using $\mathcal{L}_{freq}$ without the adaptive re-weighting $\omega$ (Eqn. 5) brings limited improvements as the frequency constraint to $\mathcal{L}_{spa}$, while employing $\omega$ further enhances performances by helping the model to learn more informative frequency components. Additionally, we visualize reconstruction results in Fig. 4 to show the improvements brought by our proposed components (more results in Appendix B).

## 5.5 VERIFICATION OF A$^2$MIM DESIGN RULES

We verify whether A$^2$MIM meets the intended design rules using the same experiment settings as Sec. 5.4: (i) A$^2$MIM is generic to incorporate advanced components proposed in previous works (*e.g.*, complex decoders, advanced prediction targets). As for the decoder structure, we replace the original linear decoder with 2-layer MLP or Transformer decoders, but find limited improvements or degenerated performances (similar to SimMIM) in Tab. 6. Inspired by PVT.V2 (Wang et al., 2022),

Table 6: Analysis of the scalability A$^2$MIM for advanced components on IN-1K.

| | Module | ResNet-50 | ViT-B |
|---|---|---|---|
| | Linear | 78.8 | 82.4 |
| | 2-layer MLP | 78.8 | 82.4 |
| Decoder | 2-layer MLP (w/ DW) | **78.9** | 82.5 |
| | 2-layer Transformer | 78.6 | 82.3 |
| | 2-layer Transformer (w/ DW) | 78.8 | **82.6** |
| | RGB | 78.8 | 82.4 |
| Target | HoG Feature | **78.9** | 82.6 |
| | DINO Feature | **78.9** | **82.7** |

we introduce a depth-wise (DW) convolution layer (w/ **DW**) to the MLP decoder (adding a $5 \times 5$ DW layer in between) and the Transformer decoder (adding a $3 \times 3$ DW layer in each FFN (Wang et al., 2022)), which brings improvements compared to the linear decoder. As for the prediction target, we follow MaskFeat to change the RGB target to the HoG feature or the output feature from ViT-B/16 pre-trained 1600-epoch by DINO (Caron et al., 2021). Tab. 6 shows that using advanced targets significantly improves the performance of $A^2$MIM for both ResNet-50 and ViT-B. Therefore, we can conclude *$A^2$MIM is a generally applicable framework.* (ii) A$^2$MIM enhances occlusion robustness and middle-order interaction among patches from experiments on ImageNet-1K in Fig. A3.

## 6 CONCLUSION

In this paper, we delved deep into MIM and answered the question of what exactly is learned during MIM pre-training. We adopted multi-order interactions to study the interaction order among image patches. We discovered that MIM essentially teaches the network to learn middle-order interactions among image patches for more complex feature extraction regardless of the network architecture. Based on our findings, we further proposed a general framework A$^2$MIM that is compatible with both Transformers and CNNs for MIM tasks aiming at enhancing patch interactions during self-supervised pre-training. Besides a different mask token mechanism, we proposed a loss in the Fourier domain to better learn the middle-order interaction. Experimental results have shown that our proposed framework improves the representations learned for both CNNs and Transformers yielding superior performance than state-of-the-arts on various downstream tasks.

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

# A    DETAILS OF COMPARISON EXPERIMENTS

This section provides experimental details for Sec. 5, *e.g.,* pre-training and evaluation on ImageNet-1K and transfer learning settings on downstream tasks.

## A.1    IMAGENET-1K EXPERIMENTS

**Pre-training.**    The default settings of $A^2$MIM for ResNet-50 and ViTs are provided in Tab. A1, following SimMIM (Xie et al., 2021b). We use AdamW (Loshchilov & Hutter, 2019) optimizer with the cosine scheduler and the linear learning rate scaling rule (Goyal et al., 2020): $lr = base\_lr \times$ batchsize / 256. Similar to current MIM methods, we only use *RandomResizedCrop* with the scale of $(0.67, 1.0)$ and do not employ other complex augmentations (*e.g.,* Rand Augment (Cubuk et al., 2020), mixups (Yun et al., 2019), or stochastic depth) during pre-training. As for ViTs, we adopt Cosine decay for 100 and 300 epochs pre-training while using Step decay (the learning rate multiplied $0.1$ at 700-epoch) for 800-epoch pre-training.

**End-to-end fine-tuning.**    Our fine-tuning settings follow common practices of supervised image classification on ImageNet-1K. As shown in Tab. A2, we fine-tune pre-trained ViTs for 100 epochs using the DeiT (Touvron et al., 2021) training recipe, which employs AdamW (Loshchilov & Hutter, 2019) optimizer with the cross-entropy (CE) loss; we fine-tune pre-trained ResNet-50 for 100/300 epochs using RSB A3/A2 (Wightman et al., 2021) settings, which employs LAMB (You et al., 2020) optimizer with the binary cross-entropy (BCE) loss. Additionally, we use layer-wise learning rate decay as (Bao et al., 2022) for fine-tuning ViT models.

Table A1: ImageNet-1K $A^2$MIM pre-training settings for ResNet-50 and ViT models.

| Configuration | ResNet-50 | ViTs |
|---|---|---|
| Pre-training resolution | $224 \times 224$ | $224 \times 224$ |
| Mask patch size | $32 \times 32$ | $32 \times 32$ |
| Mask ratio | 60% | 60% |
| Optimizer | AdamW | AdamW |
| Base learning rate | $1.5 \times 10^{-4}$ | $1 \times 10^{-4}$ |
| Weight decay | 0.05 | 0.05 |
| Optimizer momentum | $\beta_1, \beta_2$=0.9, 0.999 | $\beta_1, \beta_2$=0.9, 0.999 |
| Batch size | 2048 | 2048 |
| Learning rate schedule | Cosine | Cosine / Step |
| Warmup epochs | 10 | 10 |
| RandomResizedCrop | ✓ | ✓ |
| Rand Augment | ✗ | ✗ |
| Stochastic Depth | ✗ | ✗ |
| Gradient Clipping | ✗ | max norm= 5 |

Table A2: ImageNet-1K fine-tuning recipes for ResNet-50 (RSB A2/A3) and ViTs (DeiT).

| Configuration | ViTs | ResNet-50 | |
|---|---|---|---|
| | DeiT | RSB A2 | RSB A3 |
| FT epochs | 100 | 300 | 100 |
| Training resolution | 224 | 224 | 160 |
| Testing resolution | 224 | 224 | 224 |
| Testing crop ratio | 0.875 | 0.95 | 0.95 |
| Optimizer | AdamW | LAMB | LAMB |
| Base learning rate | $2.5 \times 10^{-4}$ | $1.5 \times 10^{-3}$ | $1 \times 10^{-3}$ |
| Weight decay | 0.05 | 0.02 | 0.02 |
| Batch size | 1024 | 2048 | 2048 |
| Learning rate schedule | Cosine | Cosine | Cosine |
| Warmup epochs | 5 | 5 | 5 |
| Label smoothing $\epsilon$ | 0.1 | ✗ | ✗ |
| Stochastic depth | 0.1 | 0.05 | ✗ |
| Gradient clipping | 5.0 | ✗ | ✗ |
| Rand Augment | (9, 0.5) | (7, 0.5) | (6, 0.5) |
| Mixup alpha | 0.8 | 0.1 | 0.1 |
| CutMix alpha | 1.0 | 1.0 | 1.0 |
| Loss function | CE loss | BCE loss | BCE loss |

## A.2    OBJECT DETECTION AND SEGMENTATION ON COCO

We adopt Mask-RCNN (He et al., 2017) framework to perform transfer learning to object detection and segmentation on COCO (Lin et al., 2014) in Detectron2[1]. For evaluation on ResNet-50, we follow MoCo (He et al., 2020) and fine-tune Mask R-CNN with the pre-trained ResNet-50-C4 backbone using $2\times$ schedule (24 epochs). For evaluation of ViTs, we follow MAE (He et al., 2022), which employs the pre-trained ViT backbone and an FPN neck (Lin et al., 2017) in Mask R-CNN, and fine-tune the model using $1\times$ schedule (12 epochs). For a fair comparison, we follow (Bao et al., 2022; Xie et al., 2021b) to turn on relative position bias in ViT (Dosovitskiy et al., 2021) during both pre-training and transfer learning, initialized as zero.

## A.3    SEMANTIC SEGMENTATION ON ADE-20K

We adopt UperNet (Xiao et al., 2018) to perform transfer learning to semantic segmentation on ADE-20K and use the semantic segmentation implementation in MMSegmentation[2]. We initialize

---

[1] https://github.com/facebookresearch/detectron2
[2] https://github.com/open-mmlab/mmsegmentation

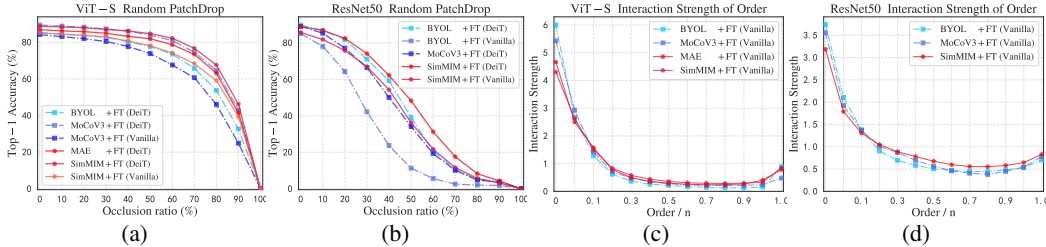

Figure A1: (a)(b): Robustness against different occlusion ratios of images (CL *vs.* MIM) is studied for both ViT-S and ResNet-50 on ImageNet-100. (c)(d): Distributions of the interaction strength $J^{(m)}$ (CL *vs.* MIM) are explored for both ViT-S and ResNet-50 on ImageNet-100. The label indicates the pre-training method + fine-tuning augmentation used, random stands for random weight initialization.

the UperNet using the pre-trained backbones (ResNet-50 or ViTs) on ImageNet-1K. For ViTs, we fine-tune end-to-end for 80K iterations by AdamW with a batch size of 16. We search a optimal layer-wise decay from {0.8, 0.9} and search optimal a learning rate from $\{1 \times 10^{-4}, 2 \times 10^{-4}, 3 \times 10^{-4}\}$ for all competitors. Similar to fine-tuning settings on COCO, we use relative position bias in ViT (Dosovitskiy et al., 2021) during both pre-training and transfer learning as (Bao et al., 2022; Xie et al., 2021b). For ResNet-50, we follow MoCo (He et al., 2020), *i.e.,* all CNN models are fine-tuned for 160K iterations by SGD with the momentum of 0.9 and a batch size of 16.

## B  EMPIRICAL EXPERIMENTS

This section provides background information and experimental details for Sec. 3. We also provide additional results of occlusion robustness evaluation and multi-order interaction strength.

### B.1  OCCLUSION ROBUSTNESS

In Sec. 3.1, we analyze robustness against occlusion of fine-tuned models on ImageNet-100 (a subset on ImageNet-1K divided by (Tian et al., 2020)) using the official implementation[3] provided by Naseer et al. (2021). Both MIM and contrastive-based methods are pre-trained 400 epochs on ImageNet-100 using their pre-training settings on ImageNet-1K. We adopt the fine-tuning training recipe as DeiT in Tab. A2 and use the same setting (100-epoch) for both ViT-S and ResNet-50. Note that we use the modified SimMIM for ResNet-50 (replacing masked patches in the input image with the RGB mean) in all experiments.

As shown in Fig. 1 and A1, we compared MIM pre-trained models supervised methods with various augmentations and contrastive learning pre-trained methods in terms of the top-1 accuracy under various occlusion ratios. We find that MIM methods show better occlusion robustness on both Transformers and CNNs. In addition to Sec. 3.1, we also provide results of salient occlusion for ViT-S and ResNet-50 on ImageNet-100 in Fig. A2. Note that the occlusion ratio means the ratio of dropped and total patches and we plot the mean of accuracy across 3 runs. We can conclude that MIM pre-trained models have stronger robustness against random and salient occlusions than supervised and contrastive-based methods.

### B.2  MULTI-ORDER INTERACTION

In Sec. 3.2, we interpret what is learned by MIM by multi-order interaction (Deng et al., 2022; Zhang et al., 2020). The interaction complexity can be represented by $I^{(m)}(i,j)$ (defined in Eqn. 1), which measures the average interaction utility between variables $i, j$ on all contexts consisting of $m$ variables. Notice that the order $m$ reflects the contextual complexity of the interaction $I^{(m)}(i,j)$. For example, a low-order interaction (*e.g.,* $m = 0.05n$) means the relatively simple collaboration between variables $i, j$, while a high-order interaction (*e.g.,* $m = 0.05n$) corresponds to the complex collaboration. As figured out in the representation bottleneck (Deng et al., 2022), deep neural networks (DNNs) are more likely to encode both low-order interactions and high-order interactions, but often fail to learn middle-order interactions. We hypothesize that MIM helps models learn more middle-order

---

[3]https://github.com/Muzammal-Naseer/Intriguing-Properties-of-Vision-Transformers

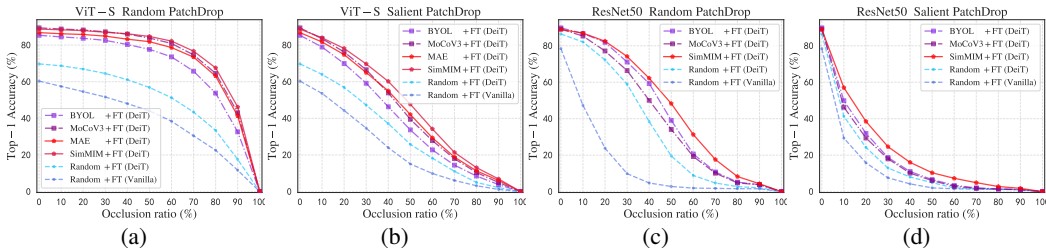

Figure A2: Robustness against various random or salient occlusion ratios of images is studied in (a)(b) for ViT-S, and (c)(d) for ResNet-50 using various experimental settings on ImageNet-100. The label indicates the pre-training method + fine-tuning setting used, random stands for random weight initialization.

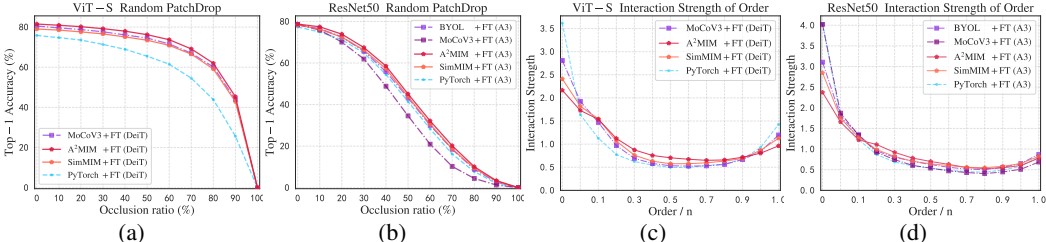

Figure A3: Verification of robustness and interaction of $A^2$MIM with ViT-S and ResNet-50 on ImageNet-1K. (a)(b): Robustness against different occlusion ratios of images is studied for $A^2$MIM and various methods. (c)(d): Distributions of the interaction strength $J^{(m)}$ are explored.

interactions since MIM has a natural advantage in cases where some parts of the image are masked out. In Fig. 1, we calculate the interaction strength $J^{(m)}$ (defined in Eqn. 2) for fine-tuned models on ImageNet-100 using the official implementation[4] provided by Deng et al. (2022). Specially, we use the image of $224 \times 224$ resolution as the input and calculate $J^{(m)}$ on $14 \times 14$ grids, *i.e.*, $n = 14 \times 14$. And we set the model output as $f(x_S) = \log \frac{P(\hat{y}=y|x_S)}{1-P(\hat{y}=y|x_S)}$ given the masked sample $x_S$, where $y$ denotes the ground-truth label and $P(\hat{y} = y|x_S)$ denotes the probability of classifying the masked sample $x_S$ to the true category.

### B.3 MIM FROM FREQUENCY PERSPECTIVE

We first plot the log magnitude of Fourier transformed feature maps of ResNet-50 with different pre-training methods using the tools[5] provided by Park & Kim (2022) on ImageNet-1K. Following (Park & Kim, 2022), we first convert feature maps into the frequency domain and represent them on the normalized frequency domain (the highest frequency components are at $\{-\pi, +\pi\}$). In Fig. 4(a), we report the amplitude ratio of high-frequency components by using $\Delta \log$ amplitude. As shown in Fig. 4(a), inpainting and MIM show similar low-pass filtering effects at convolution layers as compared to contrastive learning. This indicates that inpainting and MIM reduce noise and uncertainty induced by high-frequency features. We argue that the reconstruction performance of MIM is mainly related to low or high-order interactions of patches (Deng et al., 2022), while reconstruction performance is not directly related to the learned representation quality. Then, we provide the standard deviation of feature maps by block depth as (Park & Kim, 2022; 2021), which first calculates the feature map variance on the last two dimensions and then averages over the channel dimension for the whole dataset. Fig. 4(b) shows the feature variance of each layer of ResNet-50 with different pre-training methods on IN-1K. This figure indicates that MIM tends to reduce the feature map variance, and conversely, supervised training, inpainting, and contrastive learning based on CNN tend to increase variance. Compared to MIM, which learns better middle-order interactions, the inpainting task fails to filter out low-order interactions and thus leads to higher variance. To conclude, MIM methods learn middle-order interactions and reduce the feature map uncertainty (high frequencies) based on the CNN encoder for a generalized and stabilized feature extraction.

---

[4] https://github.com/Nebularaid2000/bottleneck
[5] https://github.com/xxxnell/how-do-vits-work

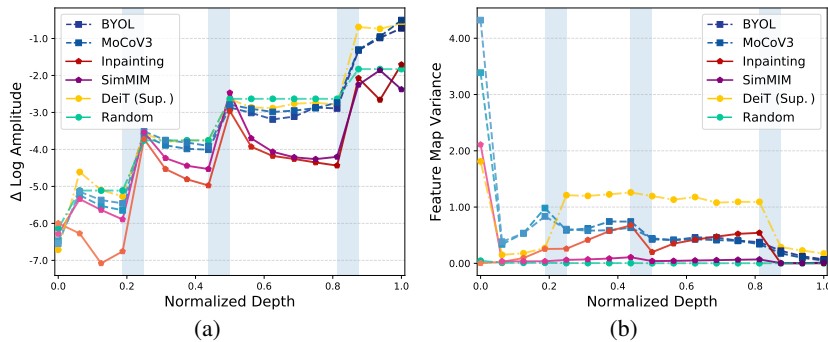

Figure A4: (a) Fourier transformed feature maps. The vertical axis is the relative log amplitudes of the high-frequency components, and the horizontal axis is the normalized depth of the network. The blue columns indicate the pooling layers, while the white columns indicate the convolution layers. (b) Feature maps variance. The vertical axis is the average variance value of feature maps. DeiT (Sup.) is supervised pre-training. The results of the randomly initialized network are plotted for reference.

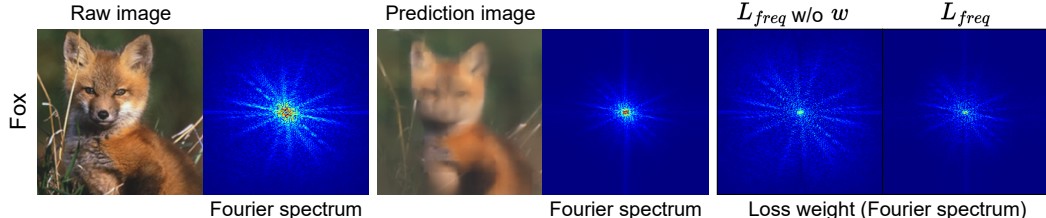

Figure A5: Visualization of predicted images and $\mathcal{L}_{freq}$ loss weight in Fourier domain. From the view of the Fourier spectrum, the raw image (left) contains 99% low-frequency components (usually present contents) and rich medium-frequency (structural patterns) and high-frequency components (local details and noises), while the predicted result (middle) provides fewer medium or high-frequency components. Calculated in the Fourier domain, the loss weights (right) of $\mathcal{L}_{freq}$ w/o $w$ help the model to learn the full spectrum while $\mathcal{L}_{freq}$ focusing on the low and medium-frequency parts, which are more likely to be low-order or middle-order interactions.

## C MORE EXPERIMENT RESULTS

### C.1 ABLATION OF THE PROPOSED MODULES

In addition to ablation studies in Sec. 5.4, we provide ablation study on the proposed $\mathcal{L}_{freq}$ in the Fourier domain, as shown in Figure A5. As we discussed in Sec. 4, we hypothesize that learning medium frequencies would help better learn middle-order interactions. we thereby propose $\mathcal{L}_{freq}$ to tackle the dilemma of $\mathcal{L}_{spa}$, which tends to learn low-frequency components (*i.e.,* contents reflected by high-order interactions). Although the reconstruction loss in the Fourier domain has a global perception, the high-frequency components are usually constructed by local details and noises (*i.e.,* low-order interactions), which might hurt the generalization abilities. Therefore, we introduce the reweight $w(u, v)$ to force the model to learn more medium-frequency components, which are identical to middle-order interactions. Then, we perform further analysis of the masked patch size for A$^2$MIM in Tab. A3. Note that we pre-train ResNet-50 for 100 epochs and ViT-B for 400 epochs on ImageNet-1K and report the fine-tuning results. As shown in Tab. A3, when the mask ratio is 60%, the optimal masked patch size is $32 \times 32$ for A$^2$MIM, which is the same as SimMIM.

Table A3: Ablation of masked patch size for A$^2$MIM based on ResNet-50 and ViT-B on ImageNet-1K.

| Model | Masked patch size | Mask ratio | PT epoch | Top-1 Accuracy (%) |
|---|---|---|---|---|
| ResNet-50 | 8 / 16 / 32 / 64 | 0.6 | 100 | 78.2 / 78.6 / **78.8** / 78.7 |
| ViT-B | 8 / 16 / 32 / 64 | 0.6 | 400 | 82.9 / 83.4 / **83.5** / 83.3 |

## C.2 Analysis Occlusion Robustness and Interaction of A$^2$MIM

We further analyze occlusion robustness and interaction strength of A$^2$MIM with ViT-S (pre-training 400-epoch) and ResNet-50 (pre-training 100-epoch) on ImageNet-1K, as shown in Fig. A3. Fig. 3(a) and 3(b) shows that A$^2$MIM is more robust to occlusion than the baseline SimMIM and contrastive learning methods with both Transformers and CNNs. Meanwhile, we find that MIM methods learn more balanced interaction strength than both supervised and contrastive learning methods in Fig. 3(c) and 3(d). A$^2$MIM further improves SimMIM by capturing more middle-order interactions ($0.2n$ to $0.6n$) with Transformers and CNNs. Therefore, we can conclude that A$^2$MIM helps the model to learn better middle-order interactions between patches for more generalized visual representation.

## C.3 Scaling-up A$^2$MIM

Additionally, we scale up the model size of backbone encoders to verify the performance of A$^2$MIM with ResNet and ViT on ImageNet-1K. As shown in Table A4, our proposed A$^2$MIM and its advanced variant A$^2$MIM+ consistently improve both the contrastive-based and MIM methods on all scale architectures, *e.g.,* A$^2$MIM outperforms SimMIM by 0.5%/0.5%/0.5%/0.2% and 0.6%/0.4% based on ViT-S/B/L/H and ResNet-50/101, demonstrating that A$^2$MIM is an architecture-agnostic and salable framework for MIM pre-training.

Table A4: ImageNet-1K fine-tuning (FT) top-1 accuracy (%) with ResNet (R) and ViT of various model scales. We adopt the 100-epoch fine-tuning protocols for both architectures.

| Methods | Supervision | ViT-S | ViT-B | ViT-L | ViT-H | R-50 | R-101 |
|---|---|---|---|---|---|---|---|
| Sup. | Label | 79.9 | 81.8 | 82.6 | 83.1 | 78.1 | 79.8 |
| MoCoV3 | CL | 81.4 | 83.2 | 84.1 | - | 78.7 | - |
| DINO | CL | 81.5 | 83.6 | - | - | 78.7 | - |
| MAE | RGB | - | 83.6 | 85.9 | 86.9 | 77.1 | - |
| SimMIM | RGB | 81.7 | 83.8 | 85.6 | 86.8 | 78.2 | 80.0 |
| MaskFeat | HoG | - | 84.0 | 85.7 | - | 78.4 | - |
| A$^2$MIM | RGB | 82.2 | 84.2 | 86.1 | 87.0 | 78.8 | 80.4 |
| A$^2$MIM+ | HoG | **82.4** | **84.5** | **86.3** | **87.1** | **78.9** | **80.5** |

## D  VISUALIZATION EXPERIMENTAL DETAILS

In addition to visualization results in Sec. 5.4, we visualize more reconstruction results of A²MIM here. Similar to Fig. 4, we ablate the proposed components in A²MIM based on ResNet-50 in Fig. A6, which demonstrates that A²MIM helps ResNet-50 learn more spatial details, *i.e.,* more middle-order interactions. Moreover, we study the effects of the mask token in both ViTs and CNNs in Fig. A7.

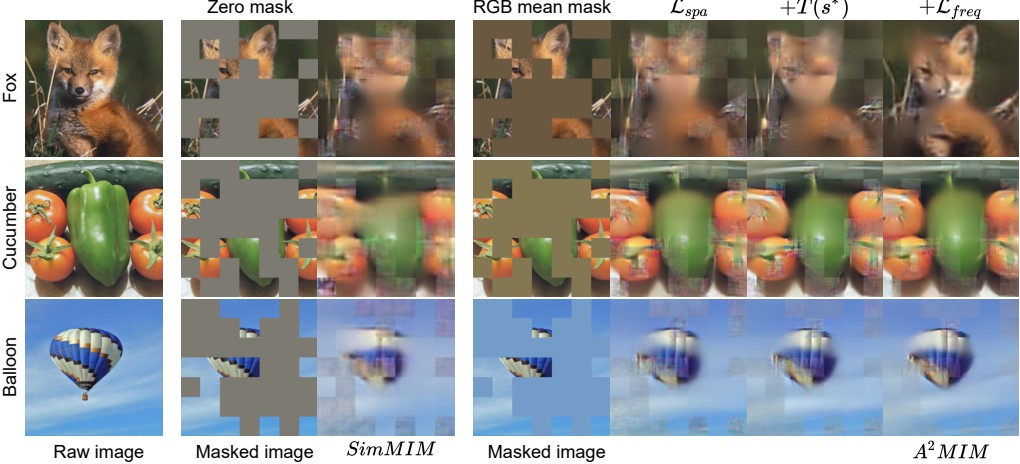

Figure A6: Visualizations of predicted results from SimMIM (middle) and our A²MIM (right) based on ResNet-50 pre-trained 100-epochs on ImageNet-1K. Notice that $T(s^*)$ denotes the mask token $T$ to the optimal stage-s in ResNet-50. We ablate the proposed components by adding them to the baseline SimMIM: replacing the zero mask with the RGB mean mask (the modified SimMIM baseline) and adding the mask token $T(s^*)$ relieve grid-like artifacts in predicted results; adding the proposed $\mathcal{L}_{freq}$ helps the model to capture more informative details.

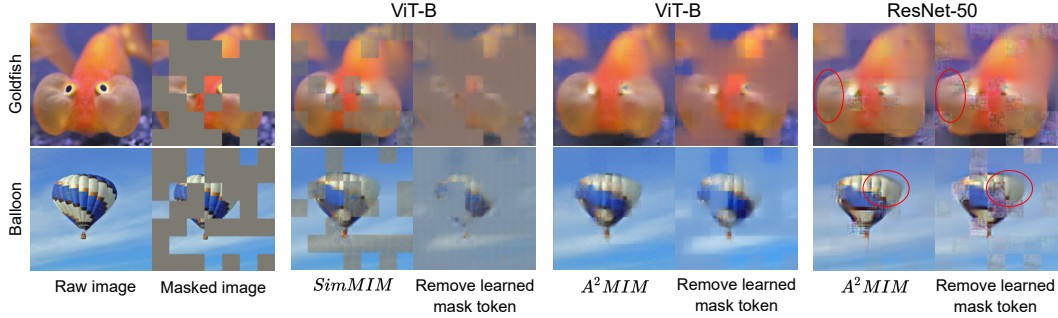

Figure A7: Visualizations of predicted results with and without the mask token on ImageNet-1K. Notice that mask tokens are adopted in the pre-trained models based on ViT-S (300-epoch) or ResNet-50 (100-epoch). Based on ViT-S, removing the mask token corrupts both contents of masked patches and overall colors in SimMIM while only corrupting the masked contents in A²MIM. Based on ResNet-50, removing the mask token slightly affects spatial details in the masked patches and causes grid-like artifacts in the unmasked patches. The different effects of the mask token in ViT-S and ResNet-50 might be because the two architectures use different spatial-mixing operators and normalization layers. As for ViTs, the self-attention operation captures informative details from unmasked patches, but the non-overlap patch embedding and layer normalization mask each patch isolated. The mask token learns the mean templates (contents) of masked patches and gathers spatial details from unmasked patches by the self-attention operation. As for CNNs, each patch shares the same contents extracted by batch normalization layers, and the convolution operation extract features from unmasked and masked patches equally. The mask token learns more high-frequency and informative details.

