# OpenReview forum: "Architecture-Agnostic Masked Image Modeling -- From ViT back to CNN"
_ICLR.cc/2023/Conference — Submitted to ICLR 2023_

### Official Review · Reviewer_nEcw · 2022-10-23

**Confidence:** 4
**Correctness:** 2
**Technical Novelty And Significance:** 3
**Empirical Novelty And Significance:** 3
**Recommendation:** 5

**Clarity, Quality, Novelty And Reproducibility:**

+ **Clarity**: The paper is well written, however, I do not agree with several claims (see Weaknesses).
+ **Quality**: Comprehensive experiments are conducted, however, the premises should be verified, i.e., do we need MIM on CNNs?
+ **Novelty**: Applying MIM to CNNs is new.
+ **Reproducibility**: Implementation details are provided.

**Strength And Weaknesses:**

### Strengths
1. The paper is well-organized and easy to follow.
2. The figures, tables, and visualizations indeed help to understand the author's point of view.
3. It is new to apply MIM on convolution neural networks.

### Weaknesses
1. I have the most concern about the motivation for this work. As we know, ViTs lack the capture of inductive bias and MIM solves this issue to some extent by enforcing visual context reasoning. However, CNNs are different, they are good at capturing inductive bias. Furthermore, as observed in the authors' results (Tab. 1), a lot of effort has been done to do MIM pre-training, however, the final results are almost the same as supervised counterparts. So is it a false proposition to apply MIM to CNNs?
2. I have another concern about the authors' declaration that MIM enables the network with a better feature extraction ability. As a common observation in previous MIM works, the linearly probing results of MIM-pretrained models are unsatisfactory, worse than the contrastive methods. MIM is known to provide transferable model parameters rather than extracting out-of-the-box features.
3. In Sec. 3.1, the authors analyze different augmentation methods, however, the setup is not fair. MIM is employed for pretraining while the others (e.g., CutMix) are for fine-tuning.
4. There are many components in the introduced method, including the mean RGB replacement, the intermediate mask, the frequency reconstruction targets and the HOG targets. How much does each component affect?

**Summary Of The Paper:**

The paper analyzes the essence of MIM and introduces a universal MIM method that can be applied to both CNNs and Transformers. There are several designs that together build the introduced method, including the mean RGB replacement, the intermediate mask, the frequency reconstruction targets and the HOG targets. Experimental results on both ResNet-50 and ViTs are shown.

**Summary Of The Review:**

I tend to weakly reject this paper in the initial comments due to the unfounded premises. I am looking forward to the authors' opinions in the rebuttal period on why we need MIM for CNNs.

---

> ### Author Response · Authors · 2022-11-14
> **Response to Reviewer nEcW (2/2)**
>
> > **Q3: In Sec. 3.1, the authors analyze different augmentation methods, however, the setup is not fair. MIM is employed for pre-training, while the others (e.g., CutMix) are for fine-tuning.**
>
> **R**: In Sec. 3.1, the robustness test is conducted under the following settings: (i) random weight initialization with no image augmentation applied; (ii) random weight initialization with different image augmentations applied; (iii) MIM pre-training as weight initialization with and without image augmentations applied. With these settings, we could see how MIM and different image augmentations affect robustness. For instance, comparing (i) and (ii), we could see how different image augmentations affect the robustness against occlusion. From (ii) itself, we could see how image augmentations compare to each other. Comparing (i) and (iii), we could see how MIM affect robustness with no image augmentation applied. Similarly, more comparisons could be made comparing different settings. The purpose is to show that MIM as pre-training improves robustness to a large extent than data augmentations.
>
>
> > **Q4: There are many components in the introduced method, including the mean RGB replacement, the intermediate mask, the frequency reconstruction targets and the HOG targets. How much does each component affect?**
>
> **R**: The ablation study of our proposed components is provided in Table 5 and 6 of the original paper. The ablation study of "mean RGB replacement, the intermediate mask, the frequency reconstruction targets" are provided in Table 5. As observed, each component individually brings performance gain and combing them further improves performance. Additional ablation studies, including the "the HOG target" is shown in Table 6.
>
> # References
>
> [1] Geirhos, Robert, et al. "ImageNet-trained CNNs are biased towards texture; increasing shape bias improves accuracy and robustness." In ICLR (2019).
>
> [2] Shikhar, Tuli, et al. "Are Convolutional Neural Networks or Transformers more like human vision?" In CogSci (2021).

---

> ### Author Response · Authors · 2022-11-14
> **Response to Reviewer nEcW (1/2)**
>
> We thank Reviewer yVqw for acknowledging that our proposed $A^2$MIM is novel and paper is well-organized and easy to follow.
>
> # Reply to Comments
>
> > **Q1: I have the most concern about the motivation for this work. As we know, ViTs lack the capture of inductive bias, and MIM solves this issue to some extent by enforcing visual context reasoning. However, CNNs are different, and they are good at capturing inductive bias. Furthermore, as observed in the authors' results (Tab. 1), a lot of effort has been done to do MIM pre-training, however, the final results are almost the same as supervised counterparts. So is it a false proposition to apply MIM to CNNs?**
>
> **R**: We agree with Reviewer nEcW that CNNs are good at capturing inductive bias. Consequently, CNNs are strongly **biased towards recognizing textures rather than shapes** [1, 2]. As we discussed in Sec. 3, MIM improves middle-order interactions among patches (shape, contour), which is exactly the weakness of CNNS. Thus, CNNs could also benefit from MIM pre-training. Also, ViTs and CNNs tend to capture low-order and high-order interactions rather than middle-order interactions meaning that both ViTs and CNNs could benefit from MIM pre-training.
>
> Reviewer nEcW also mentioned that the performance of MIM on CNN is almost the same as supervised learning: Indeed, PyTorch supervision baseline performs better on ImageNet-1K, especially on fast fine-tuning protocols like RSB A3. This is because supervised learning captures discriminative features and thus converges faster. However, for longer fine-tuning protocols like RSB A2 and other downstream tasks on COCO and ADE20K, MIM pre-training performs way better than supervised training. For instance, A$^2$MIM vs. Sup. (top-1 accuracy 80.4 vs. 79.8) under RSB A2 fine-tuning protocol, A$^2$MIM vs. Sup. (AP$^{box}$ 39.8 vs. 38.2) on COCO object detection task, and A$^2$MIM vs. Sup. (mIoU 38.3 vs. 36.1) on ADE20K semantic segmentation task. Also, we provide the experimental result on ConvNeXt-T to further show the importance of MIM pre-training on CNNs and the superiority of A$^2$MIM. In the following table, we pre-train 100 epochs under RSB A3 fine-tuning protocol, the performance of A$^2$MIM outperforms both supervised training and contrastive learning.
>
> | Methods | Supervision | ConvNeXt-T |
> |------|:-----:|:------:|
> | Sup. | Label | 79.5 |
> | BYOL | CL | 79.7 |
> | Data2Vec | Momentum | 79.0 |
> | MAE | RGB | 78.8 |
> | SimMIM | RGB | 79.3 |
> | A$^2$MIM | RGB | **79.8** |
>
>
> > **Q2: I have another concern about the authors' declaration that MIM enables the network with a better feature extraction ability. As a common observation in previous MIM works, the linearly probing results of MIM-pretrained models are unsatisfactory, worse than the contrastive methods. MIM is known to provide transferable model parameters rather than extracting out-of-the-box features.**
>
> **R**: The natural clustering property induced by pulling near similar representations and pushing away dissimilar samples makes the learned representations of contrastive-based approaches linearly separable. Linear probing is thereby commonly adopted to evaluate the performances of contrastive-based methods. Better linear separability doesn't necessarily mean better feature extraction ability. Pre-training aims to learn generic representations that are transferable to various downstream tasks. Experimental results show that under the fine-tuning protocol, MIM pre-trained models indeed outperform contrastive-based methods. Thus, we stated that "MIM enables the network with a better feature extraction ability". By "better", we mean more generalized. We will revise the wording for clarity.

---

### Official Review · Reviewer_yVqw · 2022-10-24

**Confidence:** 4
**Correctness:** 3
**Technical Novelty And Significance:** 3
**Empirical Novelty And Significance:** 3
**Recommendation:** 8

**Clarity, Quality, Novelty And Reproducibility:**

The quality, clarity and originality are sound.


**Strength And Weaknesses:**

### Strength
1. This paper provides a thoroughgoing study of MIM, and highlights the essence of MIM pre-training is to help models learn middle-order interactions between patches.
2. Based on the study of middle-order interactions of MIM, this paper proposes a novel approach that works well for both ConvNets as well as ViTs. What's more, I appreciate the management of technical details in this paper, such as ``Filling Masked Tokens with RGB Mean'' is technically sound.
3. Extensive experiments on several downstream visual recognition tasks are conducted to verify the effectiveness of the proposed approach.

### Weaknesses
1. The 2nd paragraph's name of Sec. 2 Related Work is inaccurate.``Autoregressive Modeling'' means that the output depends linearly on its own *previous* values (e.g, GPTs), while masked modeling is *bidirectional* modeling.
2. I don't see a strong connection between Sec. 3.1 and the proposed approach.
3. The scalability of A$^2$MIM is unknown, which is a crucial property for pre-training.

**Summary Of The Paper:**

This paper inspects MIM pre-training approaches and finds MIM essentially helps the model to learn better middle-order interactions between patches. Motivated by that, this paper proposes a novel MIM-based method, dubbed A$^2$MIM, that works well for both (small-sized) ConvNets & ViTs. Extensive experiments are conducted to verify the effectiveness of the proposed approach.

**Summary Of The Review:**

Overall, I think this is a solid and well-written paper.

---

> ### Author Response · Authors · 2022-11-14
> **Response to Reviewer yVqw**
>
> We thank Reviewer yVqw for acknowledging that our exploration of MIM working principle is thoroughgoing and that our proposed $A^2$MIM is novel and technically solid.
>
> # Reply to Comments
>
> > **Q1: The 2nd paragraph's name of Sec. 2 Related Work is inaccurate. "Autoregressive Modeling" means that the output depends linearly on its own previous values (e.g, GPTs), while masked modeling is bidirectional modeling.**
>
> **R**: We agree with yVqw that in GPTs, the "bidirectional modeling" and "autoregressive modeling" are two different objectives. Following GPT, iGPT [1] also explored both the autoregressive and BERT (bidirectional modeling) objectives. However, since MAE [2], SimMIM [3] and MaskFeat [4] etc., "autoregressive modeling" is used more broadly to describe the techniques where a portion of the input image is randomly masked out and then reconstructed via the pre-text task. In the computer vision community, we also followed the naming convention of these works to use "autoregressive modeling".
>
> > **Q2: I don't see a strong connection between Sec. 3.1 and the proposed approach.**
>
> **R**: Please refer to the general response **Connection between Middle-order Interactions and AMIM framework** for a better explanation of the link between Sec. 3.1 and the proposed approach. Also, the manuscript has been revised to better address this issue.
>
> > **Q3: The scalability of AMIM is unknown, which is a crucial property for pre-training.**
>
> **R**: We have done pre-training experiments based on ViTs with scaling up model sizes on ImageNet-1K. As shown in the following Table, A$^2$MIM can achieve decent performance gains with various scales of ViTs and ResNets backbones compared to existing methods, e.g., A$^2$MIM improves MIM baseline SimMIM by 0.5\%/0.5\%/0.5\%/0.2\% and 0.6\%/0.4\% based on ViT-S/B/L/H and ResNet-50/101.
>
> | Methods | Supervision | ViT-S | ViT-B | ViT-L | ViT-H | R-50 | R-101 |
> |---|:---:|:---:|:---:|:---:|:---:|:---:|:---:|
> | Sup. | Label | 79.9 | 81.8 | 82.6 | 83.1 | 78.1 | 79.8 |
> | MoCoV3 | CL | 81.4 | 83.2 | 84.1 | - | 78.7 | - |
> | DINO | CL | 81.5 | 83.6 | - | - | 78.7 | - |
> | MAE | RGB | - | 83.6 | 85.9 | 86.9 | 77.1 | - |
> | SimMIM | RGB | 81.7 | 83.8 | 85.6 | 86.8 | 78.2 | 80.0 |
> | MaskFeat | HoG | - | 84.0 | 85.7 | - | 78.4 | - |
> | A$^2$MIM | RGB | *82.2* | *84.2* | *86.1* | *87.0* | *78.8* | *80.4* |
> | A$^2$MIM+ | HoG | **82.4** | **84.5** | **86.3** | **87.1** | **78.9** | **80.5** |
>
>
> # References
>
> [1] Chen, Mark, et al. "Generative pretraining from pixels." in ICML (2020).
>
> [2] He, Kaiming, et al. "Masked autoencoders are scalable vision learners." In CVPR (2022).
>
> [3] Xie, Zhenda, et al. "SimMIM: A simple framework for masked image modeling." In CVPR (2022).
>
> [4] Wei, Chen, et al. "Masked feature prediction for self-supervised visual pre-training." In CVPR (2022).

---

### Official Review · Reviewer_dEJJ · 2022-10-24

**Confidence:** 5
**Correctness:** 2
**Technical Novelty And Significance:** 1
**Empirical Novelty And Significance:** 2
**Recommendation:** 3

**Clarity, Quality, Novelty And Reproducibility:**

This paper is clearly presented, however, the originality of this work is slightly insufficient.

**Strength And Weaknesses:**

Strengths:

    - MIM is an interesting problem/framework to study, and understanding how MIM-based methods perform well in the vision domain is important for this field.

    - The proposed method is clear to understand and is easy to follow by other researchers.

    - The experiments are extensive on ImageNet-1K, COCO, and ADE20K datasets.

Weaknesses:

   - Some statements are incorrect or even a little bit over-claimed, such as:

1)	In the abstract, the authors stated: “MIM primarily works for the Transformer family but is incompatible with CNNs.” And in the introduction section, this paper stated "To the best of our knowledge, we are the first to carry out MIM on CNNs that outperforms contrastive learning counterparts." This is not true as ConvNext [1] is adequate to handle MIM learning strategy. And this paper with CNN is actually similar to it when using patchified images with CNN.

[1] Liu, Zhuang, Hanzi Mao, Chao-Yuan Wu, Christoph Feichtenhofer, Trevor Darrell, and Saining Xie. "A convnet for the 2020s." In Proceedings of the IEEE/CVF Conference on Computer Vision and Pattern Recognition, pp. 11976-11986. 2022.

2)	“Based on this fact, we propose an Architecture-Agnostic Masked Image Modeling framework (A2MIM), which is compatible with both Transformers and CNNs in a unified way.” The proposed framework is basically similar to the regular MIM approach with trivial or not significant modifications (use RGB Mean as the Masked Tokens and add mask token on the intermediate feature map). Thus, the called "a unified way" seems incorrect. The used Fourier/Frequency domain is also not related to the core framework in this paper and can straightforwardly be applied in the vanilla MIM models.

3)	The authors stated that the proposed framework has “(a) No complex or non-generic designs are adopted to ensure compatibility with all network architectures. (b) Better middle order interactions between patches for more generalized feature extraction.” I think this is because the proposed framework is too close to the original MIM architecture.

4)	The authors claimed they “delved deep into MIM and answered the question of what exactly is learned during MIM pre-training.” This statement is very strong, however, after reading this paper carefully, I still did not get the insights or intuition about what MIM pre-training learns from the input data from the paper’s descriptions. Proper explorations and explanations are necessary to support this claim.

 - The performance in this paper is not competitive. As shown in Tables 1, 2, 4, etc., the improvement is fairly marginal (0%~0.2%). Considering that this paper used extra Fourier/Frequency domain reconstruction supervision/loss, I’m not sure whether the proposed strategy is truly effective or not.

 - The writing and organization of this paper can be improved. The used Fourier/Frequency domain reconstruction loss seems not related to the key approach of the architecture-agnostic masked image modeling framework, since it can also be applied to the regular MIM frameworks. Moreover, the insights of using this additional supervision are not clearly expressed. This part seems fragmented from others in the method.


**Summary Of The Paper:**

This paper studied the problem that MIM is compatible with the Transformer family but is incompatible with CNNs. To this end, it proposed an Architecture-Agnostic Masked Image Modeling framework (A2MIM) that is compatible with both Transformers and CNNs in a unified way. Specifically, this paper used RGB Mean as masked tokens and added mask tokens on the intermediate feature maps. A frequency domain reconstruction loss is also applied to improve the ability of learned models. Experiments are conducted on ImageNet-1K, downstream COCO detection and segmentation, and ADE20K dataset with ViT models and ResNet-50.

**Summary Of The Review:**

This paper conducted sufficient experiments. However, the method itself is not novel with marginal improvement, the novelty and originality of the method are basically not very strong. Also, many statements in this paper are slightly over-claimed (details please refer to "Strength And Weaknesses"). Thus, I tend to reject.

---

> ### Author Response · Authors · 2022-11-14
> **Response to Reviewer dEJJ (3/3)**
>
> > **Q3: “The writing and organization of this paper can be improved. The used Fourier/Frequency domain reconstruction loss seems not related to the key approach of the architecture-agnostic masked image modeling framework, since it can also be applied to the regular MIM frameworks. Moreover, the insights of using this additional supervision are not clearly expressed. This part seems fragmented from others in the method.**
>
> **R**: Please refer to the general response **Connection between Middle-order Interactions and AMIM framework** for a better explanation of the link between middle-order interactions and the proposed approach. Also, the manuscript has been revised to better address this issue.
>
> # References
>
> [1] Liu, Zhuang, et al. "A convnet for the 2020s." In CVPR (2022).
>
> [2] Zhou, Jinghao, et al. "ibot: Image bert pre-training with online tokenizer." In ICLR (2022).
>
> [3] El-Nouby, Alaaeldin, et al. "Are Large-scale Datasets Necessary for Self-Supervised Pre-training?." arXiv preprint arXiv:2112.10740 (2021).
>
> [4] Dong, Xiaoyi, et al. "Peco: Perceptual codebook for bert pre-training of vision transformers." arXiv preprint arXiv:2111.12710 (2021).
>
> [5] Fang, Yuxin, et al. "Corrupted image modeling for self-supervised visual pre-training." arXiv preprint arXiv:2202.03382 (2022).
>
> [6] He, Kaiming, et al. "Masked autoencoders are scalable vision learners." In CVPR (2022).
>
> [7] Chen, Xiaokang, et al. "Context autoencoder for self-supervised representation learning." arXiv preprint arXiv:2202.03026 (2022).
>
> [8] Xie, Zhenda, et al. "SimMIM: A simple framework for masked image modeling." In CVPR (2022).
>
> [9] Wei, Chen, et al. "Masked feature prediction for self-supervised visual pre-training." In CVPR (2022).

---

> ### Author Response · Authors · 2022-11-14
> **Response to Reviewer dEJJ (2/3)**
>
> > **Q1-3: “The authors stated that the proposed framework has “(a) No complex or non-generic designs are adopted to ensure compatibility with all network architectures. (b) Better middle order interactions between patches for more generalized feature extraction.” I think this is because the proposed framework is too close to the original MIM architecture.**
>
> **R**: The underlying idea of MIM is simple: a portion of the input image is randomly masked out and then reconstructed via the pre-text task. MAE, SimMIM [8], MaskFeat [9], and our proposed $A^2$MIM all share the same working mechanism with different designs. The designs differ in the structure of the backbone encoder, masking strategy, and even prediction target. For example, Based on MAE, MaskFeat changes the prediction target from RGB value to HOG feature with sufficient evidence to show the motivation of such design and is an acknowledged work by the community. Despite different designs, MAE, SimMIM and MaskFeat are mainly focused on the reconstruction performance during pre-training. We showed that middle-order interactions among patches are crucial for MIM during pre-training with concrete evidence in Sec. 3. We then proposed three corresponding designs in terms of masking strategy, location of masking and a loss function based on our findings. We also demonstrated the effectiveness of our approach with extensive experiments.
>
> > **Q1-4: “The authors claimed they “delved deep into MIM and answered the question of what exactly is learned during MIM pre-training.” This statement is very strong, however, after reading this paper carefully, I still did not get the insights or intuition about what MIM pre-training learns from the input data from the paper’s descriptions. Proper explorations and explanations are necessary to support this claim.**
>
> **R**: We conducted extensive experiments in Sec. 3 to support our findings that the essence of MIM pre-training is to help models learn middle-order interactions among patches which is acknowledged by other reviewers. We first verified that both MIM as a pre-training task and masking-based data augmentations enhance the robustness of networks towards occlusion via an occlusion robustness test. Then we hypothesized that MIM explicitly forces the model to learn the interactions between patches to reconstruct missing patches with high masking ratios. We performed extensive experiments on multi-order interactions to verify this hypothesis. Experimental results show that MIM pre-training enhances middle-order interactions among patches for both ViTs and CNNs. To this end, we design a novel masking strategy and loss term in the Fourier domain to enhance middle-order interaction among patches. We hope that this will help Reviewer dEJJ to understand our motivation better.
>
> > **Q2: “The performance in this paper is not competitive. As shown in Tables 1, 2, 4, etc., the improvement is fairly marginal (0%~0.2%). Considering that this paper used extra Fourier/Frequency domain reconstruction supervision/loss, I’m not sure whether the proposed strategy is truly effective or not.**
>
> **R**: Our primary contribution is not to boost performance but to design a **generic MIM framework** for all architectures by improving middle-order interactions of patches. Both our proposed novel masking strategy and loss term in the frequency domain are plug-and-play modules for all CNNs and ViTs. Compared to $A^2$MIM, iBOT [2], SplitMask [3], PeCo [4] and CIM [5] introduce additional **pre-trained tokenizer**; MAE [6] and CAE [7] require **complex transformer decoder**; iBOT and SplitMask incorporate contrastive learning. These **explicit designs** bring huge performance gains. Despite **not adopting any** of the above-mentioned **explicit designs**, $A^2$MIM still outperforms other very up-to-date SOTA methods. Our effort in making a fair comparison is also **acknowledged by Reviwer RicF**. Adopting these **explicit designs** would surely improve our performance; however, this would go against our original intention of proposing a generic MIM framework. Moreover, we also provide additional comparison results with CL and MIM baselines with scaling architecture on ImageNet-1K. As shown in the table, A$^2$MIM improves MIM baseline SimMIM by 0.5\%/0.5\%/0.5\%/0.2\% and 0.6\%/0.4\% based on ViT-S/B/L/H and ResNet-50/101 and outperforms CL baseline DINO by 0.7\%/0.6\% and 0.1\% based on ViT-S/B and ResNet-50.
>
> | Methods | Supervision | ViT-S | ViT-B | ViT-L | ViT-H | R-50 | R-101 |
> |---|:---:|:---:|:---:|:---:|:---:|:---:|:---:|
> | Sup. | Label | 79.9 | 81.8 | 82.6 | 83.1 | 78.1 | 79.8 |
> | MoCoV3 | CL | 81.4 | 83.2 | 84.1 | - | 78.7 | - |
> | DINO | CL | 81.5 | 83.6 | - | - | 78.7 | - |
> | MAE | RGB | - | 83.6 | 85.9 | 86.9 | 77.1 | - |
> | SimMIM | RGB | 81.7 | 83.8 | 85.6 | 86.8 | 78.2 | 80.0 |
> | A$^2$MIM | RGB | **82.2** | **84.2** | **86.1** | **87.0** | **78.8** | **80.4** |

---

> ### Author Response · Authors · 2022-11-14
> **Response to Reviewer dEJJ (1/3)**
>
> # Reply to Comments
>
> > **Q1-1: In the abstract, the authors stated: “MIM primarily works for the Transformer family but is incompatible with CNNs.” And in the introduction section, this paper stated, "To the best of our knowledge, we are the first to carry out MIM on CNNs that outperforms contrastive learning counterparts." This is not true as ConvNext [1] is adequate to handle MIM learning strategy. And this paper with CNN is actually similar to it when using patchified images with CNN.**
>
> **R**: ConvNeXt itself is an attempt to merge Transformer and CNN structures to come with a **new backbone structure** but has nothing to do with MIM pre-training. ConvNeXt optimized the **micro design of ResNet** with reference to the Swin Transformer. Our proposed A$^2$MIM is a **generic framework** for MIM **regardless of the actual backbone structure**. This means that A$^2$MIM could be applied to any backbone (ViTs, CNNs), including ConvNeXt. In Sec. 3.2 of the ConvNeXt paper, the pre-training on ImageNet-22K is performed with label supervision. Next, we give a comparison result of A$^2$MIM vs existing MIM algorithms with ConvNeXt-T as the backbone with 100 epochs under the RSB A3 fine-tuning protocol. As seen from the table, although ConvNeXt adopts Transformer-like micro designs, directly applying existing MIM algorithms on ConvNeXt-T yields underperforming performance. Our proposed A$^2$MIM yields the best performance, outperforming the contrastive learning baseline BYOL. Lastly, we must clarify that our proposed A$^2$MIM does not change the design of the backbone but modifies the input (as the RGB mean) and enhances middle-order interactions among patches during pre-training. There is no image patchifying operation for CNNs unless the backbone is designed to do so (like ConvNeXt).
>
> | Methods | Supervision | ConvNeXt-T |
> |------|:-----:|:------:|
> | Sup. | Label | 79.5 |
> | BYOL | CL | 79.7 |
> | Data2Vec | Momentum | 79.0 |
> | MAE | RGB | 78.8 |
> | SimMIM | RGB | 79.3 |
> | A$^2$MIM | RGB | **79.8** |
>
> > **Q1-2: “Based on this fact, we propose an Architecture-Agnostic Masked Image Modeling framework (A2MIM), which is compatible with both Transformers and CNNs in a unified way.” The proposed framework is basically similar to the regular MIM approach with trivial or not significant modifications (use RGB Mean as the Masked Tokens and add mask token on the intermediate feature map). Thus, the called "a unified way" seems incorrect. The used Fourier/Frequency domain is also not related to the core framework in this paper and can straightforwardly be applied in the vanilla MIM models.**
>
> **R**: Using RGB mean as the mask tokens and adding mask tokens on the intermediate feature map along with our proposed Fourier domain loss are all generic plug-and-play modules. iBOT [2], SplitMask [3], PeCo [4] and CIM [5] introduce additional **pre-trained tokenizer**; MAE [6] and CAE [7] require **complex transformer decoder**; iBOT and SplitMask incorporate contrastive learning. Compared to the mentioned approaches, the proposed designs of $A^2$MIM are simple yet effective. Reviewer yVqw appreciates our designs stating that "filling mask token with RGB mean is technically sound". Simplicity ensures that our proposed framework is "compatible with both Transformers and CNNs in a unified way". As shown in Table 1-4, the effectiveness of $A^2$MIM is validated by showing that $A^2$MIM outperforms both contrastive and MIM-based pre-training approaches without adopting **explicit and complex designs**. The fact that "The used Fourier/Frequency domain can straightforwardly be applied in the vanilla MIM models." mentioned by Reviewer dEJJ exactly proves that $A^2$MIM is a generic MIM framework.

---

### Official Review · Reviewer_RicF · 2022-10-25

**Confidence:** 4
**Correctness:** 3
**Technical Novelty And Significance:** 2
**Empirical Novelty And Significance:** 3
**Recommendation:** 5

**Clarity, Quality, Novelty And Reproducibility:**

**[missing discussions that may weaken the novelty]**

In the Related Work section, the authors refer to the work CIM [3].
This is an Electra-like [4] method that fills the corrupted pixels via a small network instead of mean color values, and thus [4] claimed they are the first to demonstrates that both ViT and CNN can learn rich visual representations using a unified, non-Siamese framework.
Authors are encouraged to discuss A$^2$MIM and CIM in more detail, and to update some of the corresponding descriptions.
Although this would weaken the novelty of A$^2$MIM, the related works deserve to be discussed fairly and pertinently.


**[Reproducibility]**

I believe one can easily reproduce the main results in this work given the detailed experimental configurations and source codes.


------------------
[3] Fang, Yuxin, et al. "Corrupted image modeling for self-supervised visual pre-training." arXiv preprint arXiv:2202.03382 (2022).

[4] Clark, Kevin, et al. "Electra: Pre-training text encoders as discriminators rather than generators." arXiv preprint arXiv:2003.10555 (2020).

**Strength And Weaknesses:**

**[strengths]**
- [S1] This work presents a *fresh* perspective to rethink the self-supervised learning in computer vision, say, the *middle-order* interaction. The "interaction strength" mentioned in the paper is a useful indicator to show how good the model is at modeling long-range dependency, which is actually considered to be a valuable aspect in advancing pre-training for NLP [1,2].
- [S2] The authors have put considerable effort into how to fairly compare the established algorithms, which is appreciated and allows the hypotheses presented in their paper to be fully tested. The ablation experiments also demonstrate the validity of different components in their method.


&nbsp;


**[weakness]**

There are some parts of the article that are not very easy for the reader to follow, which could be the main weakness.
I would suggest authors to explain more about the following aspects (which I find enlightening) and eventually add them to their manuscript or appendix:
  - [W1] In the Introduction, the authors claim that "it is not straightforward to directly apply mask token to CNNs". Can we replace the masked part (e.g., 16x16x3=768 pixels) with a "mask token" (e.g., 768 learnable scalars) like BERT does? This is a straightforward way to mask. Or would it be more appropriate to describe this way as "underperforming" than "not straightforward"? (considering the mediocre performance of MAE, SimMIM, etc. in Tab. 1 of the paper)
  - [W2] "MAE takes the reorganized the unmasked input patches of 112x112 as the input", can the authors explain more on this? I wonder if a substitution like the above (768 pixels to a 768-dimensional learnable vector) or some other operation has been performed.
  - [W3] According to the analysis in Sec. 3.2, middle-order interactions seem to manifest a medium- or long-range inter-patch dependency. Why the authors say "middle-order interactions could be enhanced via guiding the network to learn features of certain frequencies"? Does "certain frequencies" refer to medium or high frequencies? Why does increasing the richness of these frequencies on feature maps can promote middle-order interactions?

[W4] In addition, there appears to be some related work that has not been adequately discussed. See the "Clarity, Quality, Novelty And Reproducibility".

&nbsp;

**[open questions]**

[O1] The methodological improvements to BERT-like pre-training in this paper seems to be incremental.
Considering that the authors propose some quantitative metrics to describe middle-order operations, is it possible to design a more principled algorithm to explicitly facilitate such middle-order interactions? I believe the insights on middle-order are valuable, but the solutions proposed in the article do not seem to fully exploit their values.

&nbsp;

------------------
[1] Jawahar, Ganesh, Benoît Sagot, and Djamé Seddah. "What Does BERT Learn about the Structure of Language?." Proceedings of the 57th Annual Meeting of the Association for Computational Linguistics. 2019.

[2] Xu, Jiacheng, et al. "Discourse-Aware Neural Extractive Text Summarization." Proceedings of the 58th Annual Meeting of the Association for Computational Linguistics. 2020.

&nbsp;


**Summary Of The Paper:**

This work proposes a variant of BERT-like pretraining method for computer vision (the so-called Masked Image Modeling). The method can generalize well to both vision transformers and CNNs, thus is described as architecture agnostic.

The main contributions of this work are:
- using the mean color values to fill the corrupted pixels, which makes the self-supervised pipeline suitable for different neural network architectures;
- giving a nice insight that the middle-order interactions are important for visual representation, and making two improvements to standard BERT algorithm to facilitate their learning (i. masking intermediate features and ii. introducing supervision in the frequency domain);
- and providing some empirical evidence that supports its validity.

&nbsp;

**Summary Of The Review:**

Overall, I find this work provides some interesting insights, but the writing and presentation of the paper leaves much to be desired.
In addition, the methodological improvements to BERT pre-training in this paper seems to be trivial and piecemeal.
I hope to understand this article better in further communication with the authors and that may influence my judgment.

---

> ### Author Response · Authors · 2022-11-14
> **Response to Reviewer RicF (2/2)**
>
> > **Q4: "In the Related Work section, the authors refer to the work CIM [2]. This is an Electra-like [3] method that fills the corrupted pixels via a small network instead of mean color values, and thus [2] claimed they are the first to demonstrate that both ViT and CNN can learn rich visual representations using a unified, non-Siamese framework. Authors are encouraged to discuss AMIM and CIM in more detail and to update some of the corresponding descriptions. Although this would weaken the novelty of AMIM, the related works deserve to be discussed fairly and pertinently.**
>
> **R**: CIM is composed of a generator and an enhancer. Although the enhancer could be of any arbitrary architecture like ViTs and CNNs, the generator adopts a **pre-trained frozen image tokenizer DALL-E** and a small trainable **BEiT**. The pre-trained tokenizer and BEiT are still of the Transfomer category and essential components of CIM, thus CIM still relies on Transformers to some extent. $A^2$MIM, on the other hand, adopts no complex or non-generic designs to ensure compatibility with all network architectures. Experiment results also show that $A^2$MIM outperforms CIM. In terms of publishing order, CIM and our proposed $A^2$MIM both published on arXiv earlier this year and thus can be considered concurrent works. Following the suggestion of Reviewer RicF, we will update the manuscript to better address the difference between $A^2$MIM and CIM.
>
> # References
>
> [1] He, Kaiming, et al. "Masked autoencoders are scalable vision learners." In CVPR (2022).
>
> [2] Fang, Yuxin, et al. "Corrupted image modeling for self-supervised visual pre-training." arXiv preprint arXiv:2202.03382 (2022).
>
> [3] Clark, Kevin, et al. "Electra: Pre-training text encoders as discriminators rather than generators." In ICLR (2020).
>
> [4] Park, Namuk, and Songkuk Kim. "How Do Vision Transformers Work?." In ICLR (2022).

---

> ### Author Response · Authors · 2022-11-14
> **Response to Reviewer RicF (1/2)**
>
> We thank Reviewer RicF for acknowledging that our proposed perspective (middle-order interactions) to rethink masked image modeling is novel and that our proposed approach is sufficiently evaluated with extensive experiments.
>
> # Reply to Comments
>
> > **Q1: In the Introduction, the authors claim that "it is not straightforward to directly apply mask token to CNNs". Can we replace the masked part (e.g., 16x16x3=768 pixels) with a "mask token" (e.g., 768 learnable scalars) like BERT does? This is a straightforward way to mask. Or would it be more appropriate to describe this way as "underperforming" than "not straightforward"? (considering the mediocre performance of MAE, SimMIM, etc., in Tab. 1 of the paper)**
>
> **R**: For Transformer based MIM approaches, it is a common practice to apply the mask token after the patch embedding layer, as mentioned by Reviewer RicF, where each patch is projected to a vector. Since the patch embedding layer is not in conventional CNNs, that's why we claimed: "not straightforward to directly apply mask token to CNNs". Indeed the word "underperforming" Reviewer RicF suggested is more appropriate than "not straightforward" in terms of avoiding ambiguity. We thank Reviewer RicF for this suggestion and will revise the manuscript to address this issue.
>
> > **Q2: "MAE takes the reorganized the unmasked input patches of 112x112 as the input", can the authors explain more on this? I wonder if a substitution like the above (768 pixels to a 768-dimensional learnable vector) or some other operation has been performed.**
>
> **R**: MAE [1] only takes the visible (unmasked) patches to the encoder, the word "reorganized" comes from the operation of random shuffling and indexing all the patches to select the visible patches. Taking ResNet-50 as the encoder, MAE randomly selects 25\% from $56\times 56$ output features of the stem as unmasked patches and takes the reorganized $28\times 28$ patches as the input of four stages in ResNet. Then, the learnable mask token is expanded and applied to the positions of 75\% masked patches and both the mask tokens and unmasked patches (output from the encoder) are concatenated and reorganized as the input of the output of the decoder to reconstruct the original image.
>
> > **Q3: "According to the analysis in Sec. 3.2, middle-order interactions seem to manifest a medium- or long-range inter-patch dependency. Why the authors say "middle-order interactions could be enhanced via guiding the network to learn features of certain frequencies"? Does "certain frequencies" refer to medium or high frequencies? Why does increasing the richness of these frequencies on feature maps can promote middle-order interactions?**
>
> **R**: As discussed in Appendix B.3, ViTs have a low-pass filtering property as the feature map after self-attention layers are mainly of low-frequency components, while CNNs, on the other hand, are high-pass filters [4]. This suggests that low-frequency signals are informative to ViTs and high-frequency signals are informative to CNNs. Low frequencies correspond to low-varying and relatively global features (contents that are still recognizable even when the image is shrunk). This explains why the reconstructed images of MIM pre-training tend to be blurry. High frequencies correspond to textures and noise (CNNs are texture-biased). Now, we know that ViTs and CNNs have certain frequency bands that they each cannot model well and that both ViTs and CNNs cannot model middle-order interactions well. We also observe that better performance from using HOG as the prediction target than RGB also suggests that improving middle-order interactions helps learn better representations. Since HOG extracts edge features from relatively medium frequencies. We hypothesize that learning medium frequencies would help better learn middle-order interactions. Thus, we proposed a loss in the Fourier domain to force the model to better capture the frequencies that the model initially ignores by a weighting term. Please note that we cannot show sufficient conditions among medium frequencies and middle-order interactions at this point. However, later in Fig. A5, we show that allowing the model to learn previously ignored frequencies (mostly the medium frequencies) could indeed help improve middle-order interactions (see detailed discussion in Appendix C1).

---

### Author Response · Authors · 2022-11-14
**General Response to All Reviewers**

We sincerely thank all reviewers for their efforts and insightful reviews of our manuscript. We are encouraged to hear that the reviewers found our work provides **interesting views or understandings of MIM pre-training** (Reviewers RicF, dEJJ, yVqW) and that they think our **methodology is novel or effective** (Reviewers RicF, yVqW, nEcW). Meanwhile, all reviewers think that our **experiments are extensive and comprehensive** to verify our methods and hypotheses and that our **presentation is easy to follow**. In response to feedback, we provide point-by-point responses below to address each reviewer’s concerns and upload a revised manuscript. The changes made in the revision are highlighted in $\color{red}{red}$. Look forward to your further reply!

# Connection between Middle-order Interactions and A$^2$MIM framework

As discussed in [1], middle-order interactions (i.e., interactions of medium complexities) are more informative and essential than low or high-order interactions for visual representations. In Sec 3, we empirically show that both ViTs and CNNs tend to encode low-order or high-order interactions while MIM pre-training helps them learn more middle-order interactions, which might explain why MIM works. A typical MIM framework contains three key components: masking strategy, encoder/decoder architecture design and prediction targets. Since we aim to design an architecture-agnostic framework, we design a novel masking strategy and prediction targets to enhance middle-order interactions for both ViTs and CNNs. In particular, we design the following three modules:

* **Filling the masked tokens' positions of the input image with RGB mean.** From the aspect of the Fourier spectrum, this design provides the direct-current component of the input image, and the network is forced to model rather medium frequencies (more informative frequency components) instead of filling the mask patches with blurry color blocks (low-frequency information).

* **Placing the mask token at the intermediate feature map where middle-order interactions occur.** We choose the medium layer (e.g., $l=8$ for 12-layer ViT-B), where the output feature contains both the semantic and spatial information and the mask token can encode interactions with the medium number of tokens explicitly.

* **Designing an architecture-agnostic loss $\mathcal{L}_{freq}$ to enforce the model learn middle-order interactions.** ViTs have a low-pass filtering property as the feature map after self-attention layers are mainly of low-frequency components, while convolutions in CNNs are usually high-pass filters [2]. Low frequencies correspond to low-varying and relatively global features (contents that are still recognizable even when the image is shrunk, i.e., high-order interactions). High frequencies correspond to textures and noise (CNNs are texture-biased and favor low-order interactions). Since ViTs and CNNs have certain frequency bands that they each cannot model well and both ViTs and CNNs cannot model middle-order interactions well. We hypothesize that learning *medium frequencies* would help the model better learn *middle-order interactions*. As shown in Fig. A5, we verify that allowing the model to learn previously ignored frequencies (mostly the medium frequencies) could indeed help improve middle-order interactions (detailed in Appendix C.1).

Overall, our proposed A$^2$MIM is a general-designed framework that learns more middle-order interactions for better MIM representations.

# References

[1] Deng, Huiqi, et al. "Discovering and explaining the representation bottleneck of dnns." In ICLR (2022).

[2] Park, Namuk, and Songkuk Kim. "How Do Vision Transformers Work?." In ICLR (2022).

---

### Author Response · Authors · 2022-11-17
**Looking Forward to Post-rebuttal Feedback**

Hope everyone is going well these days. We thank all reviewers again for their effort in reviewing our paper and their insightful and helpful comments.

Based on your questions and concerns, we have provided a point-to-point response. Given that the discussion deadline is approaching, we look forward to your replies and any possible further questions.

Best,
Authors

---

### Decision · Program_Chairs · 2023-01-20

**Decision:**

Reject

**Justification For Why Not Higher Score:**

The paper is a bit of a mixed bag and fails to build a coherent narrative and deliver clear and well-founded insights. While some interesting content may already be there, presenting it properly would take some major re-work of the paper, so I recommend rejection at this point.

**Justification For Why Not Lower Score:**

N/A

**Metareview: Summary, Strengths And Weaknesses:**

The paper proposes a pre-training approach for image models based on masked image modeling (MIM) and applies it to vision transformers and ResNets. The method works a bit better than baselines.

The reviewers are overall negative about the paper (and, unfortunately, they have not responded to the authors' rebuttals). Based on the reviews, the authors' rebuttals, and the paper itself, here are the key points about the paper:

Pros:
1. Thorough experiments with the selected architectures, including modest performance improvements over baselines and an ablation study
2. Reasonable technical improvements to MIM
3. An interesting narrative around patch interactions
4. One of the first applications of MIM to ConvNets

Cons:
1. The paper mixes a few things - ViT/ConvNet, patch interactions, performance-improving modifications to MIM - and somewhat fails to combine them convincingly or do a great job at any of them separately.
 1a. The name suggests the paper is largely about ViT/ConvNet comparison and making MIM work for ConvNets. Yet, only one ConvNet architecture is studied (ResNet-50; there are some additional results in the rebuttal, but incorporating them would be a pretty big change)
 1b. The proposed improvements to MIM do not seem to affect ConvNet more than ViT, so are orthogonal to the ViT/ConvNet question.
 1c. The analysis of patch interactions is difficult to understand, and it is difficult to see what exactly it tells us. That randomly initialized and pre-trained models have different properties is not a big surprise. Results in the appendix comparing different pre-training methods seem curious but are not discussed in the main paper. Overall, this analysis may very well be useful and interesting, but from the papers in its current form it is difficult to understand.
 1d. In particular, the connection between the analysis and the proposed modifications is not very clear - is it just a high-level intuitive connection or is there more to it? Can it be shown that the proposed modifications affect patch interactions?
2. The improvement over the SimMIM baseline is not huge <1% and often quite a bit less, depending on the setting.
3. Experiments only with a limited set of models (ViT-S/B, ResNet-50). Not critical, but results with more (and larger) models would be nice.

Overall, the paper has some merit and proposes some interesting modifications to MIM pre-training and some potentially interesting analysis, but it's a bit of a mixed bag and fails to build a coherent narrative and deliver clear and well-founded insights. While some interesting content may already be there, presenting it properly would take some major re-work of the paper, so I recommend rejection at this point, but encourage the authors to improve the paper and resubmit to a different venue.